



# A large-eddy simulation study of deep-convection initiation through the collision of two sea-breeze fronts

Shizuo Fu[1, 2], Richard Rotunno[3], Huiwen Xue[4], Jinghua Chen[5], and Xin Deng[6]

[1]Key Laboratory for Humid Subtropical Eco-Geographical Processes of the Ministry of Education, Fujian Normal University, Fuzhou, China
[2]School of Geographical Sciences, Fujian Normal University, Fuzhou, China
[3]National Center for Atmospheric Research, Boulder, Colorado
[4]Department of Atmospheric and Oceanic Sciences, School of Physics, Peking University, Beijing, China
[5]Collaborative Innovation Center on Forecast and Evaluation of Meteorological Disasters, and Key Laboratory for Aerosol–Cloud–Precipitation of the China Meteorological Administration, Nanjing University of Information Science and Technology, Nanjing, China
[6]College of Agriculture, Fujian Agriculture and Forestry University, Fuzhou, Fujian, China

*Correspondence to*: Shizuo Fu (fusz@fjnu.edu.cn)

**Abstract.** Large-eddy simulations are performed to investigate the process of deep-convection initiation (DCI) over a peninsula. In each simulation, two sea-breeze circulations develop over the two coasts. The two sea-breeze fronts move inland and collide, producing strong instability and strong updrafts near the centerline of the domain, and consequently leading to DCI. In the simulation with a maximum total heat flux over land of 700 or 500 W m$^{-2}$, DCI is accomplished through the development of three generations of convection. The first generation of convection is randomly produced through the collision of the sea-breeze fronts. The second generation of convection is produced mainly through the collision of the sea-breeze fronts, but only develops in regions where no strong downdrafts are produced by the first generation of convection. The third generation of convection mainly develops from the intersection points of the cold pools produced by the second generation of convection, and is produced through the collision between gust fronts and sea-breeze fronts. As the maximum total heat flux decreases from 700 to 500 W m$^{-2}$, both the height and strength of the sea breezes are reduced, inhibiting the forcing of the first two generations of convection. These two generations of convection therefore become weaker. The weaker second generation of convection produces shallower cold pools, reducing the forcing of the third generation, and consequently weakening the third generation of convection. As the maximum total heat flux further decreases to 300 W m$^{-2}$, only one generation of shallow convection is produced.

## 1 Introduction

Deep-convection initiation (DCI) is the process through which air parcels reach their level of free convection (LFC), and remain positively buoyant over a substantial vertical excursion (Markowski and Richardson, 2010, p. 183). DCI determines when and where deep convection forms. In terms of weather forecasting, DCI may lead to severe weather phenomena, such as strong winds, heavy precipitation, hail, and/or tornadoes (e.g., Bai et al., 2019; Zhang et al., 2019). In terms of climate



projection, DCI affects the correct representation of the diurnal cycle of deep convection (Birch et al., 2015; Wang et al., 2015), and hence affects the large-scale circulation (Marsham et al., 2013) and water budget (Birch et al., 2014).

DCI relies on the presence of certain conditions. Doswell et al. (1996) proposed three ingredients for DCI: first, the environmental temperature profile must be conditionally unstable; second, sufficient moisture must be available so the rising parcels can reach saturation; third, there must be some mechanism through which the parcels are lifted to their LFC. Although the three ingredients are necessary for DCI, they are not sufficient. Many studies have shown that deep convection failed to develop despite the presence of the three ingredients (e.g., Crook, 1996; Ziegler and Rasmussen, 1998; Arnott et al.,

2006; Markowski et al., 2006; Wakimoto and Murphey, 2010). Thus, ingredients other than those listed above need to be considered.

Previous studies have identified a fourth ingredient, i.e., the cloud must be sufficiently large, so it is less diluted by entrainment and can reach higher levels (e.g., Khairoutdinov and Randall, 2006; Boing et al., 2012; Schlemmer and Hohenegger, 2014; Feng et al., 2015). Cloud tracking shows that the cloud size near its base plays an important role in

determining the cloud size aloft (Dawe and Austin, 2012; Rousseau-Rizzi et al., 2017). We note that it is the size of the boundary-layer thermal that determines the cloud size near its base. In addition, Glenn and Krueger (2017) suggested that the merger of clouds also increases cloud size and hence invigorates the clouds.

Boundary-layer convergence zones can trigger deep convection (e.g., Wilson and Schreiber, 1986; Weckwerth and Parsons, 2006; Reif and Bluestein, 2017; Huang et al., 2019). The cold front is a type of boundary-layer convergence zone (Bluestein,

2008). Although the cold front is a synoptic-scale phenomenon, its effect can also be seen on much smaller scales. Geerts et al. (2006) found that the cold front behaves as a density current on the meso-γ scale, and produces strong updrafts near its leading edge. The dryline, which is the boundary separating the moist air from the dry air (American Meteorological Society, 2020), is another type of boundary-layer convergence zone (Bluestein, 2008). It promotes DCI by producing a deep moist layer and a deep updraft (Ziegler and Rasmussen, 1998; Miao and Geerts, 2007; Wakimoto and Murphey, 2010). The gust

front, which is the leading edge of the outflow (American Meteorological Society, 2020), is also a common type of boundary-layer convergence zone. Under suitable conditions, gust front can consecutively trigger new convection (Rotunno et al., 1988; Fu et al., 2017). In addition, compared to an isolated boundary-layer convergence zone, the synergy of multiple boundary-layer convergence zones is especially favorable for DCI (e.g., Wakimoto et al., 2006).

When synoptic-scale forcing is weak, surface heterogeneity can produce boundary-layer convergence zones (Drobinski and

Dubos, 2009), and thereby trigger deep convection (e.g., Hanesiak et al., 2004; Frye and Mote, 2010; Taylor et al., 2011; Guillod et al., 2015). Previous modelling studies revealed that DCI tends to occur over the patch of surface with a stronger sensible heat flux because stronger updrafts and higher humidity coexist over this patch (Patton et al. 2005; van Heerwaarden and de Arellano 2008; Kang and Bryan, 2011; Garcia-Garreras et al., 2011). In an idealized modelling study by Rieck et al. (2014), it was found that although deep convection is able to develop in the absence of surface heterogeneity, it

occurs later and is weaker than in the presence of surface heterogeneity.



Land-sea contrast is an important type of surface heterogeneity. It is capable of producing the sea-breeze circulation (Miller et al., 2003; Antonelli and Rotunno, 2007; Crosman and Horel, 2010). As a type of boundary-layer convergence zone, the sea-breeze front is also capable of triggering deep convection. A sea-breeze front can trigger deep convection on its own (Blanchard and Lopez, 1985). However, it mostly triggers deep convection by interacting with other boundary-layer

convergence zones, such as horizontal convective rolls (Wakimoto and Atkins, 1994; Dailey and Fovell, 1999; Fovell, 2005), gust fronts (Kingsmill, 1995; Carbone et al., 2000), or river breezes (Laird et al., 1995).

In this study, we explore a special type of land-sea contrast, i.e., a peninsula. In this situation, two sea-breeze circulations develop, with their sea-breeze fronts moving toward each other. Both observational studies (Blanchard and Lopez, 1985; Carbone et al., 2000) and numerical studies (Nicholls et al., 1991; Zhu et al., 2017) showed that the oppositely-moving sea-

breeze fronts over a peninsula can collide with each other and produce deep convection.

The processes involved in DCI occur at very small spatiotemporal scales (~1 km and ~1 min; Weckwerth, 2000; Wilson and Roberts, 2006; Soderholm et al., 2016). However, the grid interval is typically ~1 km in the aforementioned numerical studies (Nicholls et al., 1991; Zhu et al., 2017), and is not sufficient to resolve these processes. In this study, DCI is simulated with large-eddy simulations (LESs) whose spatial resolution is 100 m and output frequency is 1 min. As shown

below, simulations with such a high spatiotemporal resolution allow for a deeper understanding of DCI.

Section 2 presents the methods. Section 3 describes the general evolution of the simulated convection. Section 4 analyzes the environment before DCI. The process of DCI is presented in Sect. 5. In Sect. 6, the sensitivity to the total heat flux from the surface is discussed. Section 7 summarizes the major findings.

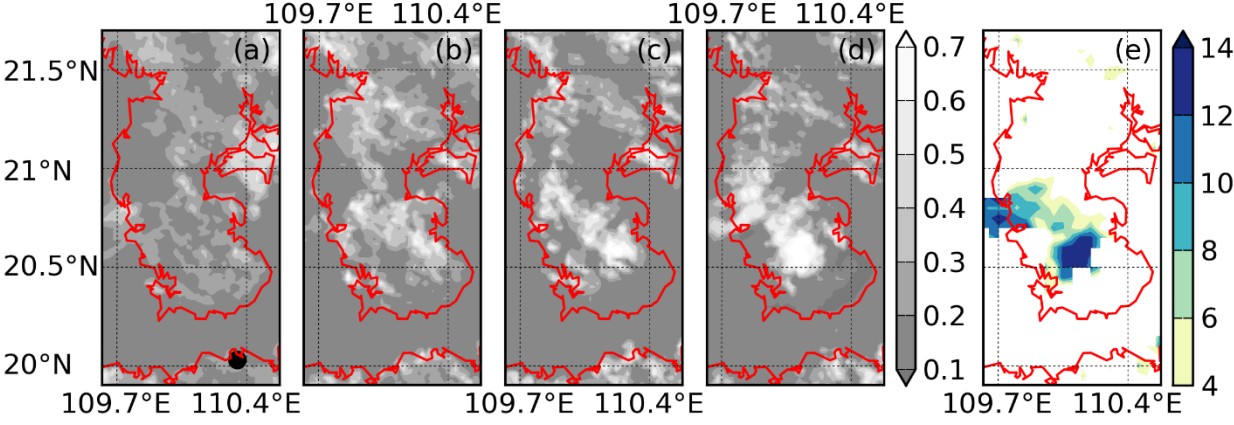

**Figure 1: Observed albedo at the wavelength of 0.47 μm and at (a) 11:00, (b) 12:00, (c) 13:00, and (d) 14:00 LST, September 25, 2018. (e) Retrieved cloud-top height (km) at 14:00 LST. The black dot in (a) indicates the location of Haikou station, where the sounding was obtained.**



## 2 Methods

### 2.1 Case

Deep convection frequently occurs over Leizhou Peninsula (Bai et al., 2020), which is the southernmost part of mainland China. A deep-convection case on 25 September 2018 was chosen for this study. Figure 1 shows the albedo and cloud-top height observed by the geosynchronous Himawari-8 satellite. At 11:00 LST (local standard time; Fig. 1a), the clouds are weak and disorganized. As time goes by (Figs. 1b and 1c), the clouds become stronger and organize themselves along the centerline of the peninsula. At 14:00 LST (Fig. 1d), an even stronger cloud forms over the southern part of the peninsula.

Figure 1e reveals that the cloud top is higher than 12 km at 14:00 LST, indicating that deep convection has formed.

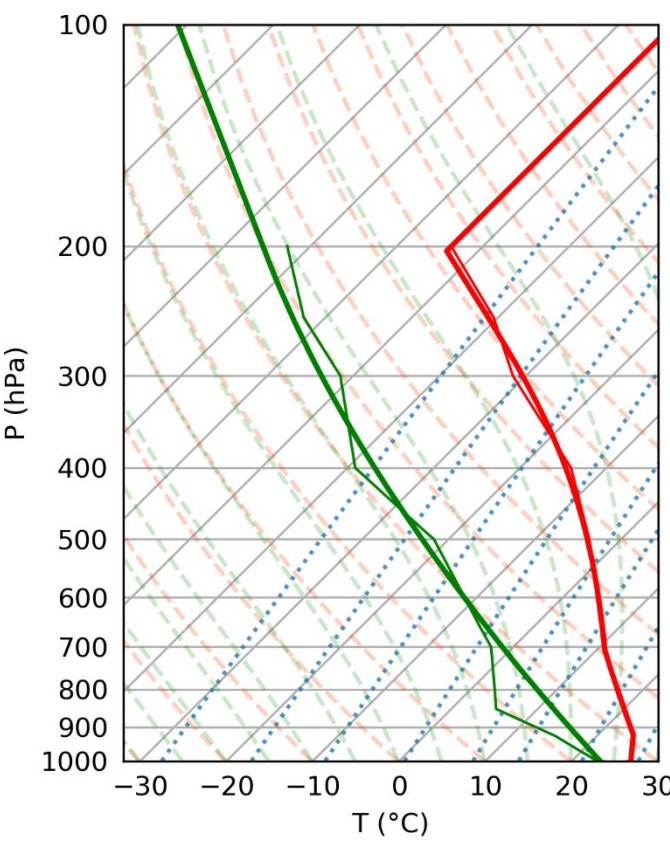

**Figure 2: The thin lines show the observed sounding from Haikou station at 08:00 LST, 25 September 2018. The thick lines show the smoothed sounding used to initialize the simulations. The red lines show the temperature profile and the green lines show the dew point temperature profile.**

Figure 2 shows the sounding at Haikou station at 08:00 LST. The location of Haikou station (110.35 ° E, 20.03 ° N) is indicated by the black dot in Fig. 1a. The thick lines in Fig. 2 show the profiles used to initialize the simulations. The initial profiles are obtained by smoothing the observed profiles, which are shown with the thin lines. In the troposphere, the initial temperature profile has a three-layer structure. From the surface to 925 hPa, the temperature profile follows a moist adiabat.



Such a layer is stable with respect to a dry adiabatic process but neutral with respect to a saturated, moist adiabatic process. This layer is probably the result of radiative cooling during the preceding night. From 925 hPa to 700 hPa, the temperature decreases with height slower than the dry adiabat but faster than the moist adiabat. Such a layer is stable with respect to a dry adiabatic process but unstable with respect to a saturated, moist adiabatic process. From 700 hPa to the top of the troposphere, the temperature profile again follows a moist adiabat. We analyzed several soundings from the same station, and found similar three-layer structures on days with deep convection. Similar three-layer structures are also seen in other

studies (e.g., Beringer et al., 2001; Shepherd et al., 2001).

The environmental wind speed is generally less than 2 m s$^{-1}$ below 700 hPa, and is approximately 4 m s$^{-1}$ between 700 hPa and 400 hPa (not shown). This is consistent with the known fact that sea-breeze circulation develops preferentially when the environmental wind is sufficiently weak (Miller et al., 2003; Crosman and Horel, 2010). In this study, the environment wind is set to zero to simplify the analysis.

**2.2 Model**

The present simulations were conducted with release 19.7 of Cloud Model 1 (CM1; Bryan and Fritsch, 2002). CM1 solves the nonhydrostatic equations using the time splitting method, which explicitly calculates the acoustic term in the horizontal while implicitly calculating the acoustic term in the vertical (Skamarock and Klemp, 1994). In this study, CM1 is configured as an LES, where the subgrid-scale turbulence is parameterized with the turbulence kinetic energy (TKE) scheme (Deardorff,

1980). Cloud microphysics is represented with the one-moment Kessler scheme. The Coriolis factor is $0.5 \times 10^{-4}$ s$^{-1}$, which is the value at 20 °N. Radiative transfer is not considered.

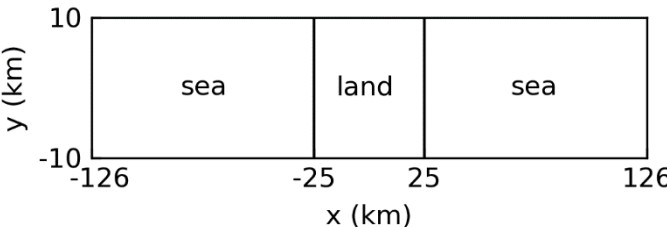

**Figure 3: Domain configuration.**

Figure 3 shows the domain configuration. A peninsula is located at the domain center. The sea is to the left and the right side

of the peninsula. The width of the peninsula is 50 km, which is approximately the width of Leizhou Peninsula (Fig. 1). The domain is 252 km in the cross-coast direction ($x$-direction hereafter), 20 km in the along-coast direction ($y$-direction hereafter), and 17 km in the vertical direction (z-direction hereafter). In the $x$-direction, tests show that a length of ~250 km is sufficient to accommodate the sea-breeze circulations, i.e., the lateral "open" boundary conditions do not affect the simulated sea-breeze circulations. In the $y$-direction, tests show that a periodicity length of 20 km is large enough to capture

the largest eddies because further increasing this length to 30 km produces similar results. The horizontal resolution is uniformly 100 m. The vertical resolution is 40 m from surface to 4 km, then gradually stretches to 200 m at 6.4 km, and then remains at 200 m up to the model top. Note that the vertical resolution in this study is even higher than that in Antonelli and





Rotunno (2007), where the sea-breeze circulation is reasonably well simulated. A Rayleigh damping layer is used above 13 km to absorb gravity waves.

The prescription of the surface fluxes is guided by the ERA5 reanalysis data. For the sea surface, the Bowen ratio is 0.1, and the total heat flux (which is the sum of sensible heat flux and latent heat flux) is constant at 100 W m$^{-2}$ during the considered time. For the land surface, the Bowen ratio is 0.2, and the total heat flux displays a prominent diurnal cycle. Therefore, the total heat flux is prescribed as

$$THF = THF_m \sin\left(2\pi \frac{t}{1\,day}\right), (1)$$

where $THF_m$ is the maximum total heat flux, and $t$ is time. On the chosen day, the maximum total heat flux over the peninsula is 480 W m$^{-2}$. However, during the whole of September of 2018, the maximum total heat flux varies between 700 and 300 W m$^{-2}$. Therefore, three values are tested here, i.e., 700, 500, and 300 W m$^{-2}$. Hereafter, the three simulations are respectively referred to as THF700, THF500, and THF300. In addition, the roughness length of the surface is set to 0.1 m for the land surface and $2 \times 10^{-4}$ m for the sea surface (Wieringa, 1993).

Each simulation is run from $t = 0$ to 12 h. The three-dimensional (3-D) fields were saved every 10 min. However, our initial analysis showed that the 10-min data could not resolve the DCI processes. Therefore, the model was restarted and the 3-D fields were saved every 1 min during the DCI period. In simulations THF700, THF500, and THF300, the 1-min data covers $t = 7.5$ to 9.5 h, $t = 8.5$ to 10.5 h, and $t = 10.5$ to 12 h.

    Online Lagrangian parcels were used to investigate the processes of DCI. Note that only the resolved-scale velocity is
considered in the calculation of the parcel trajectory, and the subgrid-scale velocity is not considered. Although including the subgrid-scale velocity may change the individual parcel trajectory (Weil et al. 2004; Yang et al. 2008), it does not change the statistics of a large number of parcel trajectories (Yang et al. 2008). The latter is the case in this study. Parcels were released at every grid point in the region of -10 km $< x <$ 10 km, -10 km $< y <$ 10 km, and 0 km $< z <$ 2 km. The total number of parcels is $2 \times 10^6$. By restarting the simulations, we can release the parcels at different times. The release times will be
mentioned in the appropriate sections.

## 3 General evolution of the convection

    Figures 4a-d show the pseudo albedo at four times in simulation THF700. The pseudo albedo is the parameterized albedo of the simulated clouds. It is calculated in order to compare the simulated clouds with the satellite-observed clouds (Zhang et al., 2005). Note that the sun rises at $t = 0$ h in model time but at 6:00 in LST, so model time + 6 h is LST. The four times in
Figs. 4a-d correspond to the four times of Figs. 1a-d. Clouds do not develop over the sea in our simulations (not shown); thus, only the land part is shown in Fig. 4. A comparison of Figs. 1 and 4 indicates that the simulation qualitatively reproduces the observed evolution of the convection. At $t = 5$ h, Fig. 4a shows that the clouds are small and disorganized. As time goes by (Figs. 4b and 4c), the cloud field shrinks, and finally becomes a line of strong convective cells near the domain



center (Fig. 4d). Figure 4e reveals that the cloud top of the strongest convective cell is higher than 12 km at $t = 8$ h. Note that

the cloud top is defined as the highest grid point with cloud water mixing ratio greater than 0.01 g kg$^{-1}$. In addition, Figs. 4a-c also indicate that the convective cells near the edges of the cloud field, which are actually the positions of the sea-breeze fronts, are bigger than those between the two edges. This point will be revisited in Sect. 4.1.

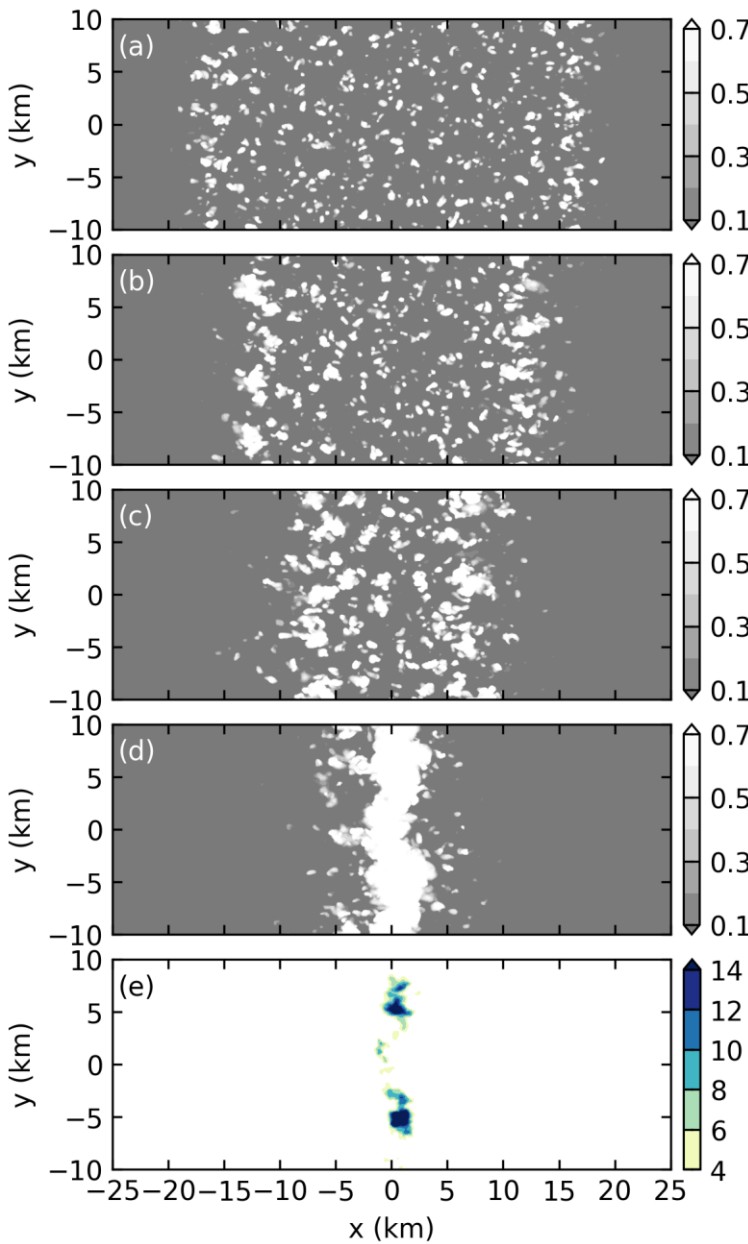

**Figure 4: Pseudo albedo in simulation THF700 at (a) $t = 5$ h, (b) $t = 6$ h, (c) $t = 7$ h, and (d) $t = 8$ h. (e) Cloud-top height (km) at $t =$**
**8 h. Note that only the land part is shown.**

The reanalysis data shows that the maximum total heat flux is 480 W m$^{-2}$ on the chosen day. However, a maximum total heat flux of 700 W m$^{-2}$ is required for the model to reproduce the observations. In simulations THF500 and THF300, the DCI is respectively delayed by 1 and 3 hours. Several reasons may explain the difference between the simulations and the observations. For example, Miller et al. (2003) pointed out that environmental wind increases the low-level convergence associated with sea-breeze front. Therefore, the omission of environmental wind may produce weaker sea-breeze fronts, and hence postpone DCI. In addition, the omission of topography, the cyclic boundary condition in the $y$-direction (i.e., the peninsula is infinite in the $y$-direction), and the simplification of coastline shape may all contribute to the difference between the simulations and the observations.

## 4 The preconvective environment

It is found that the effect of sea-breeze circulation on the preconvective environment is similar in all three simulations. Thus, only simulation THF700 is presented. As shown later, the effect of the sea-breeze circulation is different before and after the collision of sea-breeze fronts. These two periods are therefore discussed separately.

### 4.1 Before the collision of sea-breeze fronts

Figure 5 shows the $y$-averaged cross-coast wind, along-coast wind, potential temperature, vapor mixing ratio, convective available potential energy (CAPE), and convective inhibition (CIN) at $t = 6$ h in simulation THF700. In this study, CAPE and CIN are calculated by lifting a parcel that possesses the mean properties of the lowest 0.2 km. Two sea-breeze circulations develop, with the left one around the left coast and the right one around the right coast (Fig. 5a). Weak mean winds also develop in the $y$-direction due to the Coriolis effect (Fig. 5b). At $t = 6$ h, the sea-breeze fronts have moved inland by 15 km (Fig. 5a). As the sea breeze moves inland, the sensible heat flux increases its potential temperature (Fig. 5c), and the latent heat flux increases its vapor mixing ratio (Fig. 5d). Therefore, CAPE increases while CIN decreases from the coast to the sea-breeze front (Figs. 5e and 5f). Figure 5f also indicates that CIN maximizes near the coasts. As found by Cuxart et al. (2014), this is because the return flow has a downward component near the coast, which produces subsidence warming and thereby stabilizes the atmosphere.

Between the two sea-breeze fronts, Fig. 5c indicates that the potential temperature is higher than that in the sea breeze. For the air between the sea-breeze fronts, it is always heated by the stronger sensible heat flux from the land surface; while for the air in the sea breeze, it is first heated by the weaker sensible heat flux from the sea surface, and then heated by the stronger sensible heat flux from the land surface. Therefore, the air between the sea-breeze fronts receives more heat and is hence warmer than the air in the sea breeze. In addition, the large sensible heat flux between the sea-breeze fronts forces the boundary layer to grow rapidly, entraining a significant amount of dry air into the boundary layer. Therefore, the vapor mixing ratio between the sea-breeze fronts slowly decreases with time (not shown). At $t = 6$ h, the vapor mixing ratio between the sea-breeze fronts is lower than that in the sea breeze (Fig. 5d). As a result, CAPE between the sea-breeze fronts





is substantially smaller than that in the sea-breeze (Fig. 5e). Figure 5f shows that CIN has been completely removed between the sea-breeze fronts. It is worth mentioning that although CAPE is large and CIN is nearly zero over a large portion of the land, no deep convection develops at this time.

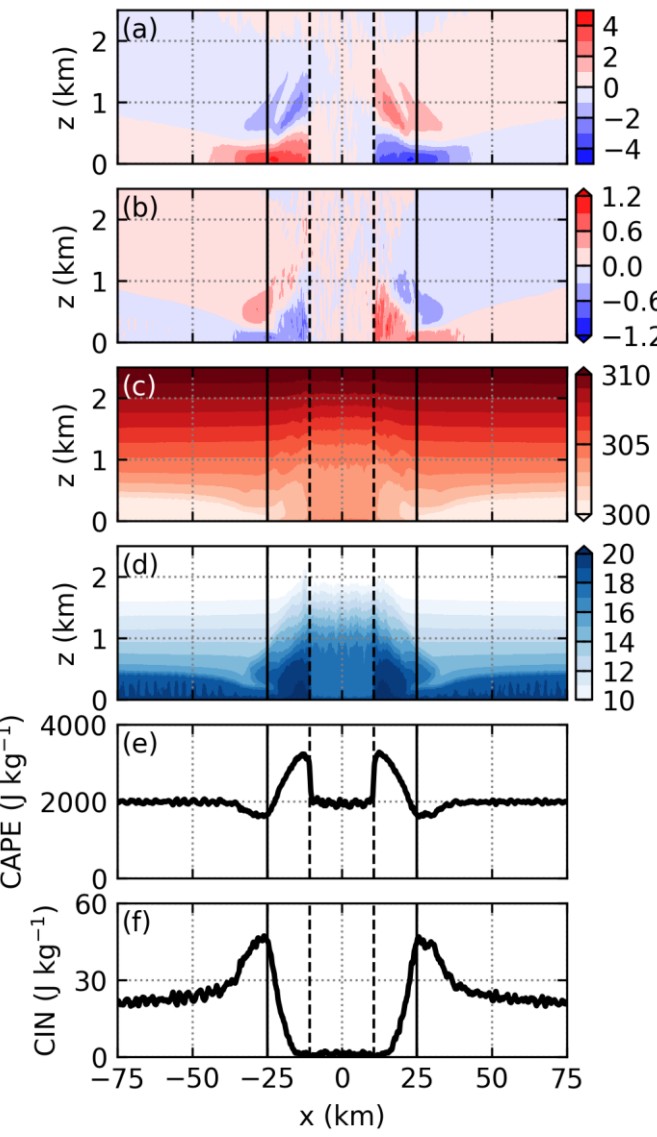

**Figure 5: y-averaged (a) cross-coast wind (m s⁻¹), (b) along-coast wind (m s⁻¹), (c) potential temperature (K), (d) vapor mixing ratio (g kg⁻¹), (e) CAPE, and (f) CIN at t = 6 h in simulation THF700. The solid lines at x = -25 and 25 km denote the coasts. The dashed lines denote the identified positions of the sea-breeze fronts.**

Thermal size plays an important role in determining DCI, as mentioned in Sect. 1. It is therefore interesting to investigate whether the thermal size is affected by the sea-breeze circulation. This is done by comparing the thermal sizes near the sea-breeze fronts to those between the sea-breeze fronts. The thermals within the sea breezes are small and weak (not shown),





and are thus not analyzed. Due to the turbulent nature of the flow, it is probably not useful to investigate the size of individual thermals. Therefore, we respectively composite the thermals near the sea-breeze fronts and those between the sea-breeze fronts, and then compare the sizes of the composite thermals.

Here we briefly present the method of compositing. The details are given in Appendix A. First, we define the positions of the sea-breeze fronts. Second, we identify thermals, and define the position of each thermal. Third, a thermal is defined as a "left-front thermal" if its distance from the left sea-breeze front is less than 1 km; a thermal is defined as a "right-front thermal" if its distance from the right sea-breeze front is less than 1 km; and a thermal is defined to be an "intermediate thermal" if it is between the two sea-breeze fronts and is more than 1 km away from each sea-breeze front. Fourth, a

procedure similar to that used by Finnigan et al. (2009) and Schmidt and Schumann (1989) is used to composite the thermals at a given output time. Note that the three types of thermals are composited separately. Finally, the size and vertical velocity of the composite thermals are respectively nondimensionalized by the boundary layer height ($z_i$) and the convective velocity scale ($w^*$), and averaged over a period of time to give the mean composite thermal.

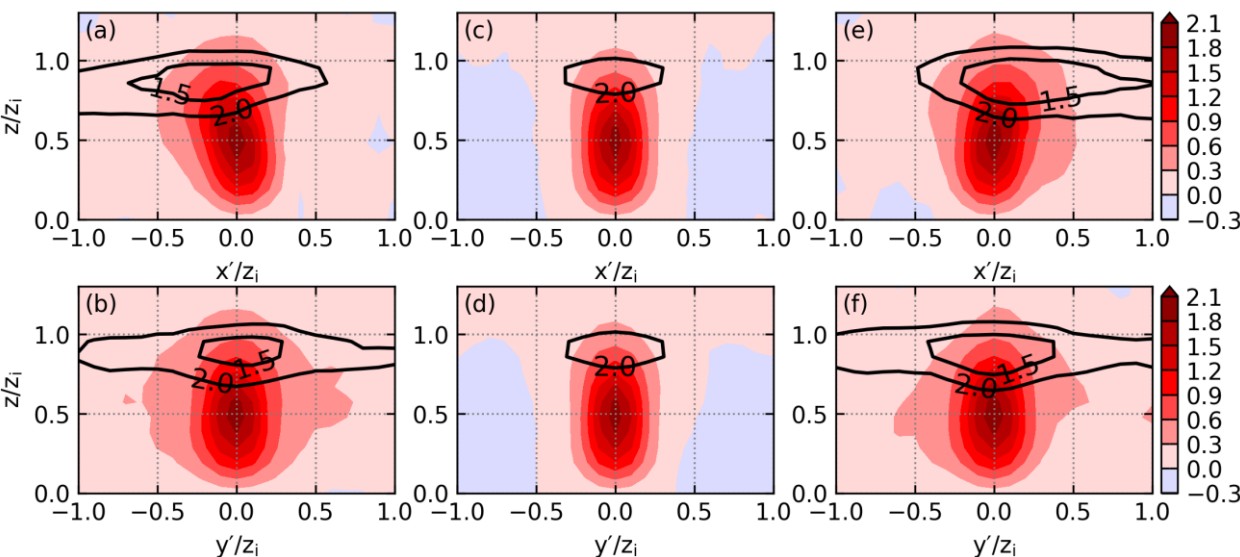

**Figure 6: (a) $x$-$z$ cross section and (b) $y$-$z$ cross section of the composite left-front thermal averaged from $t$ = 5 h 30 min to 6 h 30 min in simulation THF700. The filled contours indicate the nondimensionalized vertical velocity, and the black contours indicate the dew point depression (K). (c)-(d) are the same as (a)-(b) except for the composite intermediate thermal. (e)-(f) are the same as (a)-(b) except for the composite right-front thermal.**

Figure 6 shows the mean composite left-front thermal, mean composite intermediate thermal, and mean composite right-front thermal averaged from $t$ = 5 h 30 min to 6 h 30 min. Figure 6a indicates that the mean composite left-front thermal tilts

to the left; and Fig. 6e indicates that the mean composite right-front thermal tilts to the right. This tilting is why we composite the left-front thermal and the right-front thermal separately. Due to the convergence near the sea-breeze fronts, Fig. 6 shows that the mean composite thermals near the sea-breeze fronts are substantially larger than the mean composite intermediate thermal. Furthermore, the mean composite thermals near the sea-breeze fronts are moister than the mean



composite intermediate thermal. The larger and moister left- and right-front thermals explain the fact that the horizontal

scales of the clouds near the sea-breeze fronts are bigger than those between the two sea-breeze fronts (Fig. 4).

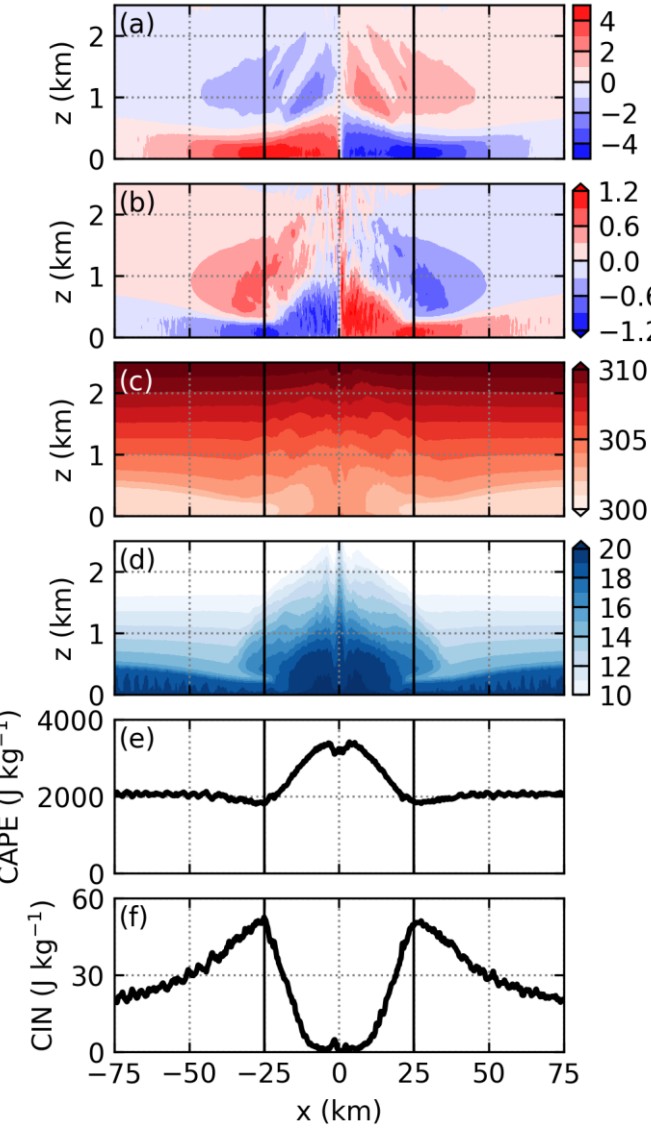

**Figure 7: The same as Fig. 5, except at** $t = 7$ **h 50 min.**

Interestingly, Fig. 6 also shows that the vertical velocity of the mean composite thermals near the sea-breeze fronts is similar

to that of the mean composite intermediate thermal. This suggests that, in the present simulation, the effect of a sea-breeze

front is to increase the size of the thermal but not to strengthen the updraft of the thermal. Many previous studies showed that

the updraft near the front of a gravity current is strengthened (e.g., Iwai et al., 2011); however, several observational studies

indicate that the magnitude of updraft is not changed by boundary-layer convergence zones (e.g., Kraus et al., 1990; Wood et





al., 1999). Hence, further studies are required to understand the effect of sea-breeze fronts on the magnitude of the thermal
updrafts.

## 4.2 After the collision of sea-breeze fronts

Figure 7 shows the $y$-averaged cross-coast wind, along-coast wind, potential temperature, vapor mixing ratio, CAPE, and
CIN at $t$ = 7 h 50 min in simulation THF700. Figure 7a shows that the two sea-breeze fronts have collided near the domain
center. The potential temperature near the domain center is almost the same as that at $t$ = 6 h (cf. Figs. 5c and 7c). Figure 7d
shows that the vapor mixing ratio near the domain center is much higher than that at $t$ = 6 h (cf. Figs. 5d and 7d) as a result
of the sea breezes transporting vapor inland. Consequently, CAPE near the domain center becomes much higher than that at
$t$ = 6 h (cf. Figs. 5e and 7e), and is greater than 3000 J kg$^{-1}$ (Fig. 7e). Figure 7f shows that the distribution of CIN at $t$ = 7 h
50 min is similar to that at $t$ = 6 h, except that the portion with near-zero CIN becomes smaller.

After the collision of sea-breeze fronts, deep convection quickly develops, which does not allow us to composite the
thermals. Figure 8a shows the horizontal cross section of vertical velocity at $z$ = 0.5$z_i$ and at $t$ = 7 h 50 min, and indicates
that updrafts exist along most of $y$-direction around $x$ = 0 km. In order to compare the vertical velocity at $t$ = 7 h 50 min to
the mean composite thermals, the length and vertical velocity are respectively nondimensionalized by $z_i$ and $w^*$ and shown
in Fig. 8b. At various places, the length of the nondimensional updraft is greater than 2 in the $x$-direction, comparable to the
mean composite thermals near the sea-breeze fronts (cf. Fig. 6). In the $y$-direction, the length of the nondimensional updraft
is much larger than that of the mean composite thermals near the sea-breeze fronts shown in Figs. 6b and 6f. Strictly
speaking, the nondimensional updraft in Fig. 8b should not be compared to the mean composite thermals in Fig. 6 because
the former is an instantaneous cross section while the latter is the mean of multiple thermals. Nevertheless, we think it is safe
to conclude that the horizontal scale of the thermals after the collision is even larger than that before the collision. Thus, due
to the higher CAPE and even bigger thermal, the environment after the collision of sea-breeze fronts is a more favorable
environment for DCI.

## 5 The processes of DCI

As shown above, the deep convective cells align along the centerline of the domain, and are mostly confined between $x$ = -2
and 2 km (Fig. 4d). Thus, the fields averaged from $x$ = -2 to 2 km include the information of all convective cells. By
analyzing the 1-min instantaneous fields averaged from $x$ = -2 to 2 km, it is clearly seen that DCI is not a one-step process;
instead, DCI is accomplished through the development of multiple generations of convection. Here, we define the first
generation of convection as the convective cells initiated solely by the sea-breeze fronts, and the second generation of
convection as the convective cells affected by the first generation of convection. Similarly, we can also define the third
generation of convection. In simulations THF700 and THF500, three generations of convection can be identified. In
simulation THF300, only one generation of convection develops. It is worth pointing out that each generation generally



contains multiple convective cells, and these convective cells may not develop synchronously. Due to this complexity,
subjectivity is unavoidable when identifying the different generations of convection. In this section, we use simulation
THF700 to show the detailed processes involved in DCI.

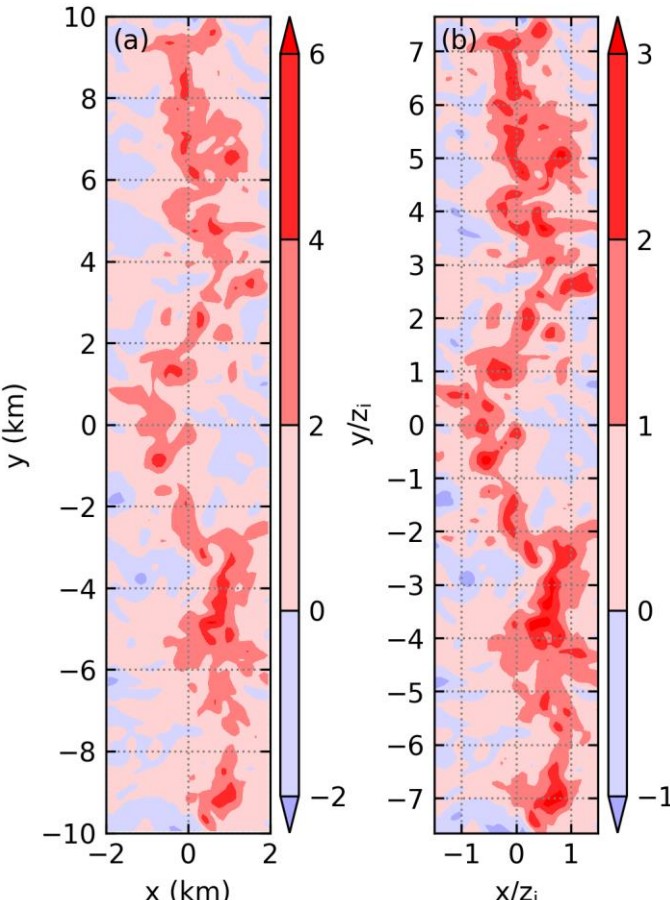

**Figure 8: Horizontal cross section of (a) dimensional vertical velocity (m s⁻¹) and (b) nondimensional vertical velocity at $z = 0.66$**
**km and $t = 7$ h 50 min in simulation THF700.**

Some definitions that are frequently used in this section are presented here. For a parcel that is initially below the LFC, it
must be lifted by some lifting mechanism before it can ascend to LFC (Sect. 1). A parcel is defined as "having been lifted" if
it is below the LFC at time $t_0$, above the LFC at time $t_0 + 1$ min and ascends by more than 0.12 km from time $t_0$ to $t_0 + 1$
min (corresponding to a mean vertical velocity of 2 m s⁻¹). We further define time $t_0$ as the time when the parcel is lifted,
and the position of the parcel at $t_0$ as the position where the parcel is lifted.

We define a grid to be within a cold pool if its surplus in density potential temperature (temperature surplus hereafter) $\Delta\theta_\rho =$
$\theta_\rho - \overline{\theta_\rho}$ satisfies

$$\Delta\theta_\rho < -1 \text{ K}. \quad (2)$$


The density potential temperature $\theta_\rho = \theta \left[ 1 + \left( \frac{1}{\varepsilon} - 1 \right) q_v - q_c - q_r \right]$, where $\theta$ is potential temperature, $q_v$ the vapor mixing

ratio, $q_c$ the cloud water mixing ratio, $q_r$ the rain water mixing ratio, $\varepsilon = R_d / R_v$ with $R_d$ and $R_v$ respectively the gas constants of dry air and vapor. The reference density potential temperature $\overline{\theta_\rho}$ is the $\theta_\rho$ averaged from $x$ = -10 to 10 km and from $y$ = -10 to 10 km at the start of the 1-min output. A column is defined to be within the cold pool if its lowest grid satisfies Eq. (2). We search upward along the column, and label the index of the first grid level not satisfying Eq. (2) as $k$, then the height of ($k$-1)-th grid is defined as the depth of the cold pool. The temperature surplus averaged over all the grids

below the $k$-th grid is defined as the mean temperature surplus of the column.

## 5.1 First generation of convection

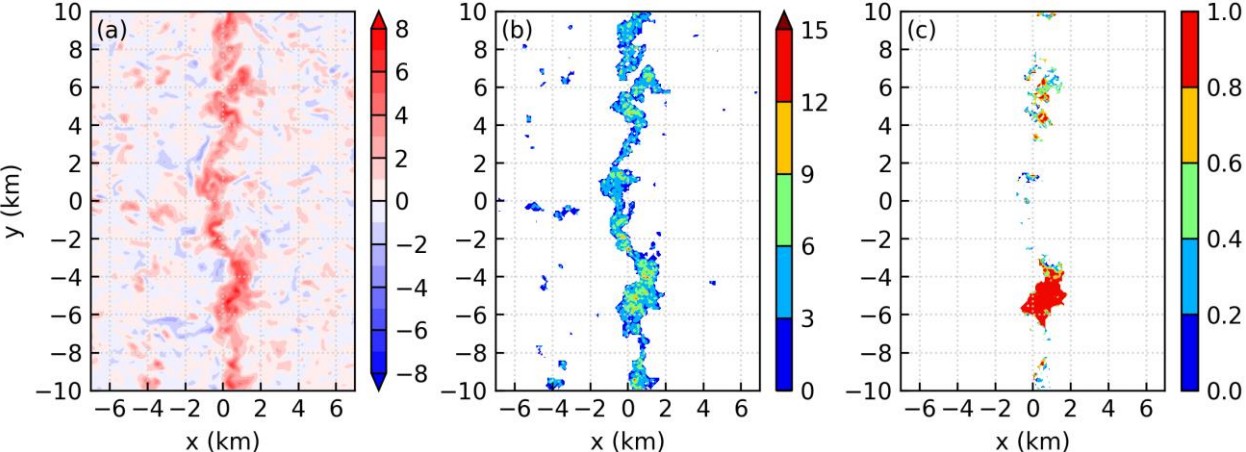

**Figure 9: (a) The vertical velocity at $z$ = 0.88 km. (b) The number of parcels that are lifted at each grid. (c) The fraction of parcels that rises above 4 km in all parcels that are lifted. The parcels are released at $t$ = 7 h 35 min and tracked forward for 20 min. The**
**results are at $t$ = 7 h 47 min of simulation THF700.**

Figure 9a shows the vertical velocity at $z$ = 0.88 km, which is the LFC averaged from $x$ = -1 to 1 km and from $y$ = -10 to 10 km right after the collision of sea-breeze fronts. Due to the collision of the oppositely-moving sea-breeze fronts, a quasi-linear updraft develops around $x$ = 0 km (Fig. 9a). Parcels are released at $t$ = 7 h 35 min and tracked forward for 20 min to investigate where the first generation of convection develops. Figure 9b shows the number of parcels that are lifted at $t$ = 7 h

47 min at each grid. Inspection of Figs. 9a and 9b shows that the positions where the parcels are lifted are almost the same as the positions where the vertical velocity is greater than 2 m s$^{-1}$. This is a direct result of the definition. Figure 9b also indicates that more parcels are lifted from regions with stronger updrafts (cf. Fig. 9a), similar to that found by Tang and Kirshbaum (2020).





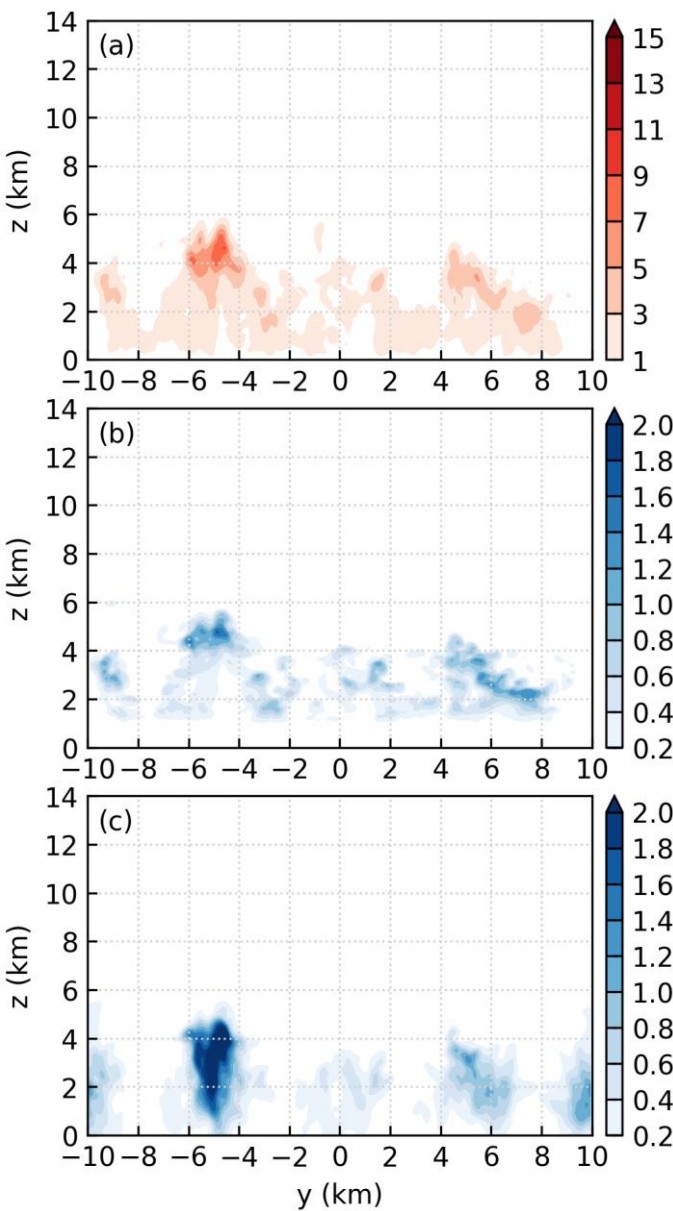

**Figure 10: (a) Vertical velocity (m s$^{-1}$), (b) cloud water mixing ratio (g kg$^{-1}$), and (c) rain water mixing ratio (g kg$^{-1}$) averaged from $x = $ -2 to 2 km at $t = $ 7 h 56 min in simulation THF700.**

A critical height is used to distinguish the parcels that rise to high levels from the parcels that do not rise to high levels. The maximum height reached by the first generation of convection is approximately 6 km, as shown below. We therefore set the critical height to be 4 km. Figure 9c shows the fraction of parcels that rise above 4 km in all the parcels that are lifted at $t = $ 7 h 47 min. Not all the parcels that are lifted manage to rise above 4 km. Furthermore, the parcels that rise above 4 km are mainly lifted from the regions where the updrafts are wider in the $x$-direction. In this simulation, the updraft is nearly





continuous in the $y$-direction (Figs. 9a and 8). In this situation, the dimension in the $y$-direction is not an appropriate measure of the updraft size. However, the dimension in the $x$-direction is an appropriate measure of the updraft size. Therefore, Fig. 9 shows that convective cells developing from larger updrafts rise to higher levels, consistent with previous studies (e.g., Dawe and Austin, 2012; Rousseau-Rizzi et al., 2017).


Figures 10a-c respectively show the vertical velocity, cloud water mixing ratio, and rain water mixing ratio averaged from $x$ = -2 to 2 km at $t$ = 7 h 56 min. Note that the mean vertical velocity is defined as $\bar{w} = \sum_{i=1}^{N} \frac{wf(w)}{N}$, where $f(w) = 1$ if $w > 0$ and $f(w) = 0$ if $w \leq 0$. $N$ is the number of grids averaged. The two convective cells around $y$ = -5 and 5 km in Fig. 10a correspond to the two clusters of parcels that rise above 4 km (cf. Fig. 9c). They are produced directly through the collision of sea-breeze fronts, and hence belong to the first generation of convection. At this time, these two convective cells have reached their mature stages and the cloud tops are close to their maximum heights. Cloud water forms as a result of the strong updrafts (Fig. 10b). Through a series of cloud microphysical processes (Fu et al., 2019), rain water forms around $y$ = -5 and 5 km (Fig. 10c).


The rain shafts around $y$ = -10, 0, and 10 km are produced by the earlier convective cells, which also belong to the first generation of convection but develop several minutes earlier than the convective cells around $y$ = -5 and 5 km. In addition, it is found that the convective cell around $y$ = -5 km is the strongest in the first generation of convection, and seems to be randomly produced. A detailed analysis indicates that the randomness is related to the variability of the sea-breeze fronts in the $y$-direction (not shown). We note that because of the cyclic boundary condition in the $y$-direction, the rain shafts around $y$ = -10 and 10 km are actually produced by the same convective cell. Similar caution should be used when interpreting other variables.



### 5.2 Second generation of convection

Figures 11a and 11b respectively show the depth and the temperature surplus of the cold pools at $t$ = 8 h. Four cold pools can be identified, corresponding to the four rain shafts in Fig. 10c. At this time, the cold pools have not yet merged, and are generally shallower than 0.2 km and warmer than -2 K near their edges.


Figures 11c and 11d respectively show the vertical velocity at $z$ = 0.2 and 0.88 km. At $z$ = 0.2 km, downdrafts exist in the interiors of the cold pools. Closed rings of updraft can be seen near the edges of the cold pools. At $z$ = 0.88 km, downdrafts are also discernable at the centers of the cold pools. However, updrafts do not form closed rings; instead, four updraft segments are produced. Figure 11d also shows that the updraft segments are separated by the downdrafts. A comparison of Figs. 11c and 11d reveals that some updrafts are deep because they are seen at both $z$ = 0.2 and 0.88 km, while the other updrafts are shallow because they are seen at $z$ = 0.2 km but not at $z$ = 0.88 km. The formation of deep and shallow updrafts will be further explored below.


Parcels are released at $t$ = 7 h 55 min and tracked forward for 25 min to investigate the development of the second generation of convection. Figure 11e shows that parcels are lifted from the regions with vertical velocity greater than 2 m s$^{-1}$





(cf. Fig. 11d), similar to that observed in Fig. 9. The second generation of convection is stronger than the first generation, as

seen below. We accordingly use a higher critical height, i.e., 6 km, to distinguish the parcels that rise to high levels from the

parcels that do not rise to high levels. The fraction of parcels that rise above 6 km is shown in Fig. 11f. Each updraft segment

produces some parcels that rise above 6 km. At the time shown in Fig. 11f, the fraction of parcels that rise above 6 km is

generally small for each updraft segment. However, the fraction will increase to over 0.8 in the following 8 minutes (not

shown).

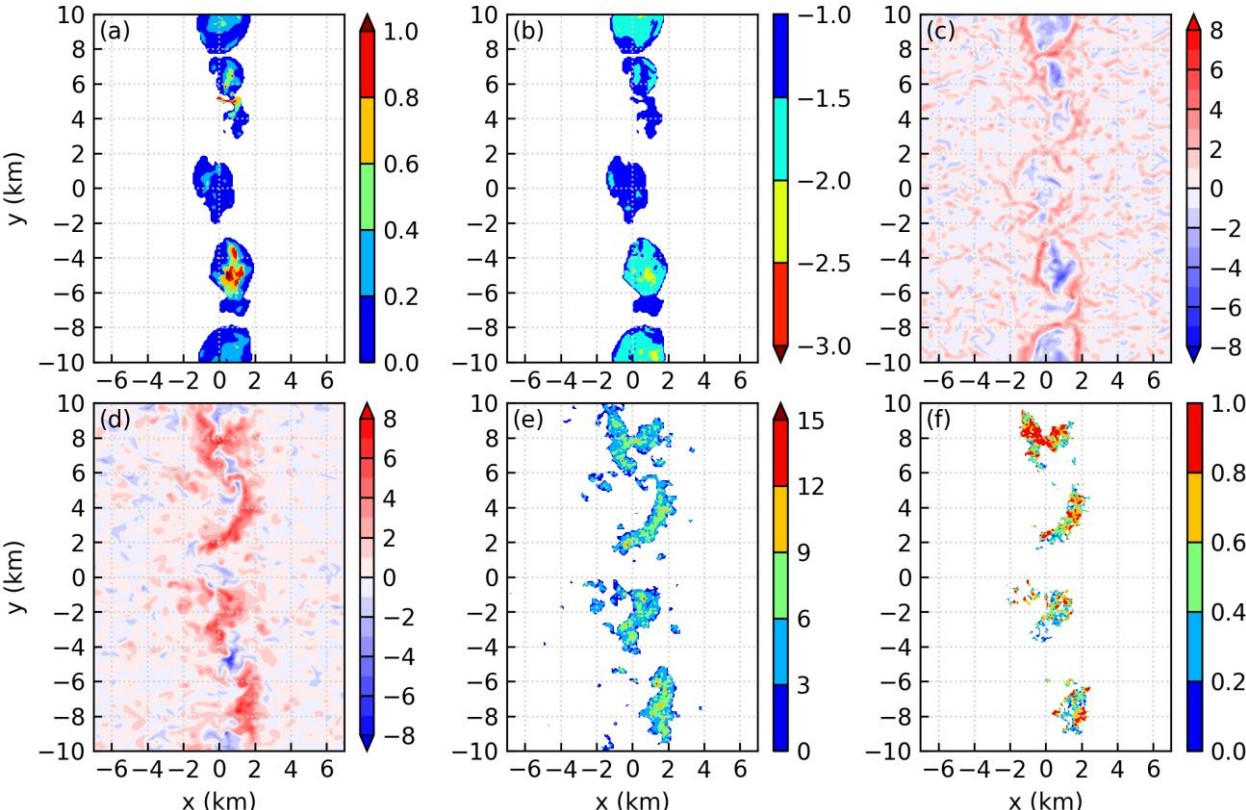


**Figure 11: (a) and (b) are respectively the depth (km) and temperature surplus (K) of the cold pools. (c) and (d) are respectively the vertical velocity at $z$ = 0.2 and 0.88 km. (e) The number of parcels that are lifted at each grid. (f) The fraction of parcels that rises above 6 km in all parcels that are lifted. The parcels are released at $t$ = 7 h 55 min and tracked forward for 25 min. The results are at $t$ = 8 h of simulation THF700.**

Figures 12a and 12c respectively show the vertical cross section of cross-coast wind and vertical velocity at $y$ = -0.95 km.

The black contour indicates the edge of the cold pool. At this time, the cold pool spans from $x$ = -0.8 to 0.8 km and is

generally shallower than 0.2 km. The sea breezes experience weak lifting when they encounter the shallow gust fronts (Fig.

12a). After being lifted, the sea breezes continue to move and collide with each other, producing a wide and deep updraft.

Obviously, the dominant forcing mechanism of the updraft is the collision of the sea-breeze fronts, rather than the collision

between the gust fronts and the sea-breeze fronts. In fact, this vertical cross section is representative of all regions where





deep updrafts are produced. In particular, at regions not covered by cold pools, the collision of sea-breeze fronts is the only

forcing mechanism that produces the updraft.

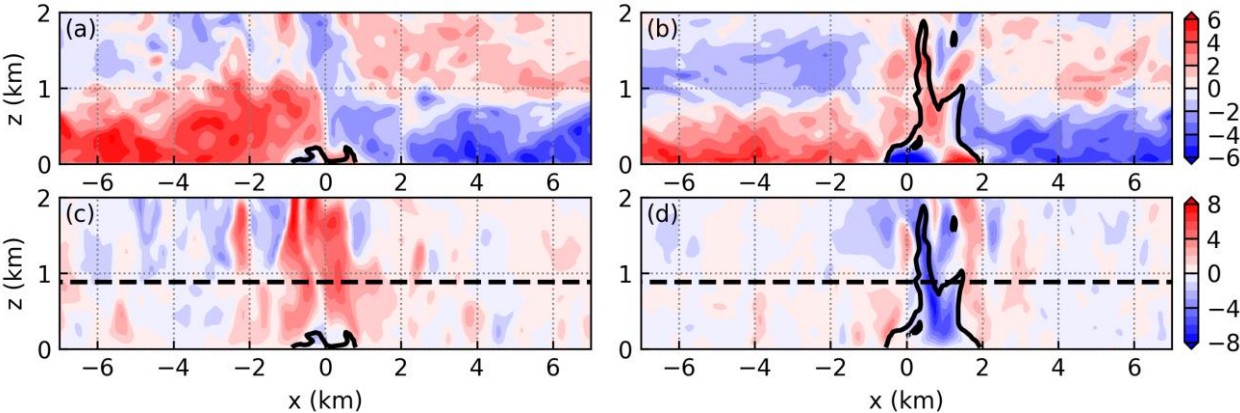

**Figure 12: Vertical cross section of (a) cross-coast wind and (c) vertical velocity at $y$ = -0.95 km. (b) and (d) are the same as (a) and (c) except at $y$ = -4.95 km. The black contour indicates the edge of the cold pool. The dashed line indicates the LFC. The results are at $t$ = 8 h of simulation THF700.**

Figures 12b and 12d show the vertical cross section at $y$ = -4.95 km. This vertical cross section is representative of the

regions where only shallow updrafts are produced. The cold pool is readily seen in Fig. 12b. The cold-pool depth near the

edge is approximately 0.2 km, which is rather shallow. The updraft forced by such a shallow gust front is also shallow (Fig.

12d). More importantly, although the sea breezes that are lifted by the gust fronts continue to move toward each other and

collide around $x$ = 1 km (Fig. 12b), deep updraft cannot be produced. This is because the downdraft from aloft suppresses

the development of updraft.

We can now summarize the processes through which the second generation of convection is produced. In the regions where

downdrafts exist, deep updrafts cannot be produced because of the suppression effect exerted by the downdrafts produced by

the first generation of convection. As a result, new convection does not develop in these regions. In the regions where

downdrafts do not exist, the collision of sea-breeze fronts produces deep and wide updrafts. The second generation of

convection therefore develops from these regions. In addition, the updrafts that produce the second generation of convection

are more organized, and are generally wider in the $x$-direction than the updrafts that produce the first generation (cf. Figs. 9a

and 11d), so the second generation of convection is stronger than the first generation.

Figures 13a-c respectively show the $y$-$z$ plots of the vertical velocity, cloud water mixing ratio, and rain water mixing ratio

averaged from $x$ = -2 to 2 km at $t$ = 8 h 13 min. Figure 13a reveals multiple convective cells, corresponding to the clusters of

parcels that rise above 6 km (cf. Fig. 11f). The animation of $y$-$z$ plots (not shown) reveals that the second generation of

convection eventually rises above 10 km. These deeper convective cells produce more cloud water (Fig. 13b), and

consequently more rain water (Fig. 13c) than the first generation of convection.





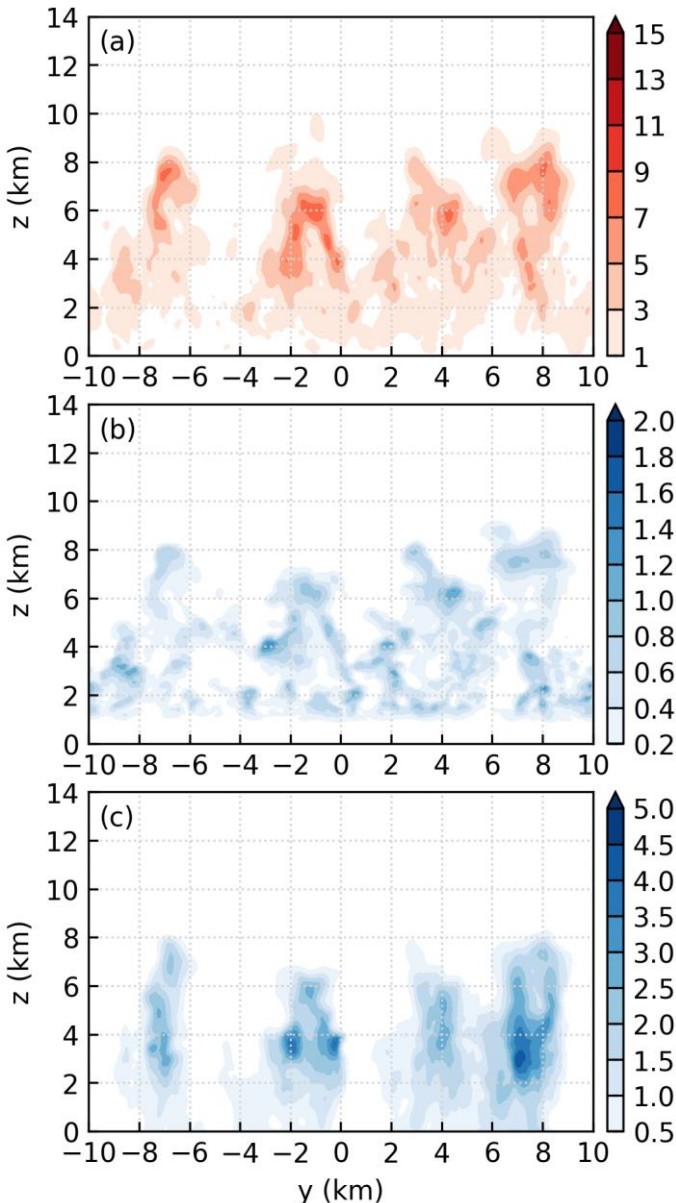


**Figure 13: The same as Fig. 10, but at $t = 8$ h 13 min.**

### 5.3 Third generation of convection

Figures 14a and 14b show the depth and the temperature surplus of the cold pool at $t = 8$ h 17 min. It reveals a single cold
pool that spans the whole $y$-direction. This cold pool is generally deeper and colder than the cold pools at $t = 8$ h (cf. Figs.
11 and 14). A careful examination of Fig. 14b reveals that the cold pool is not homogeneous, but composed of two cold
pools that are separated by the warmer regions (indicated by the blue color) around $y = 1$ and 10 km. This conclusion is





further confirmed by the inspection of the evolution of cold pools (not shown). The collision of these two cold pools produce
several intersection points. The most prominent intersection points are at $(x, y) = (2, 1)$ and $(2, 10)$ km.

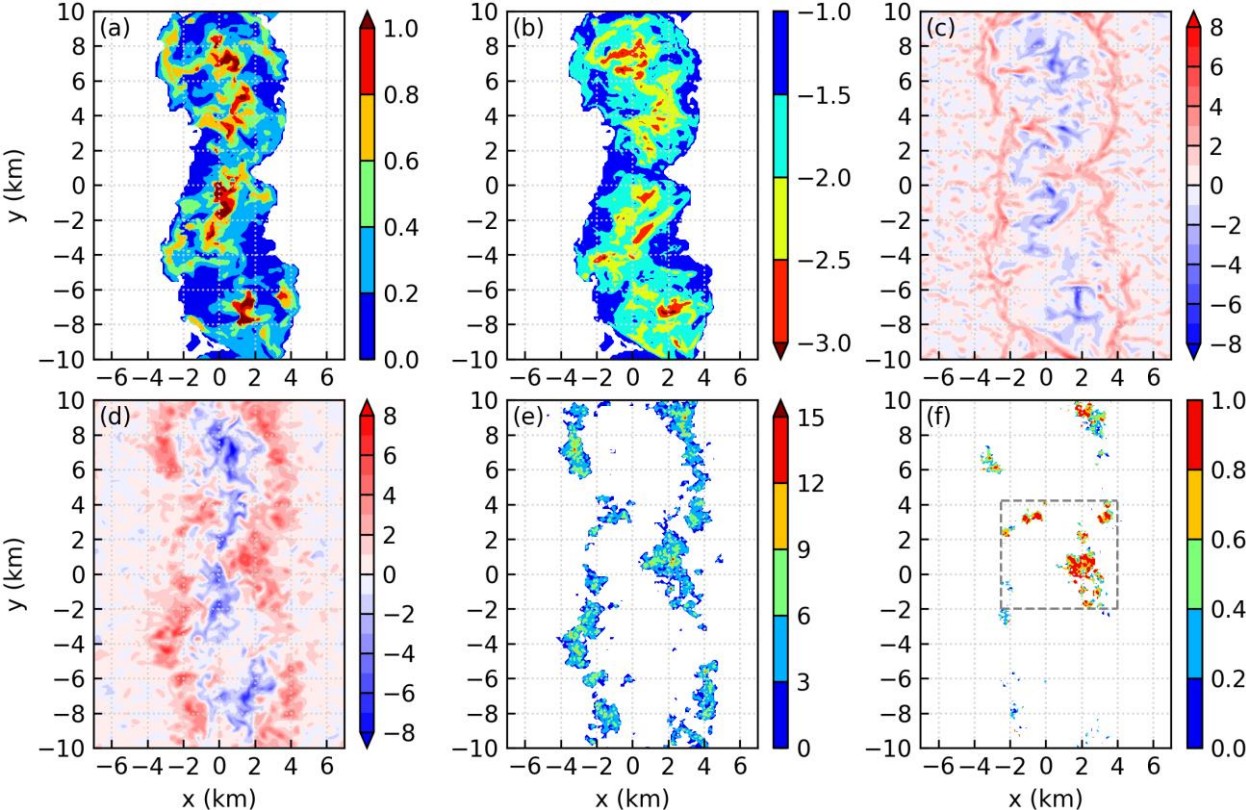

**Figure 14: (a) and (b) are respectively the depth (km) and temperature surplus (K) of the cold pools. (c) and (d) are respectively the vertical velocity at $z = 0.2$ and 0.88 km. (e) The number of parcels that are lifted at each grid. (f) The fraction of parcels that rises above 8 km in all parcels that are lifted. The grey rectangle encloses the parcel clusters that merge. The parcels are released at $t = 8$ h 5 min and tracked forward for 40 min. The results are at $t = 8$ h 17 min of simulation THF700.**

Figures 14c and 14d respectively show the vertical velocity at $z = 0.2$ and 0.88 km. Downdrafts can been seen in the interiors
of the cold pools, both at $z = 0.2$ and 0.88 km. At $z = 0.2$ km, updrafts are produced at the edges of the cold pools and are
nearly continuous. However, the updrafts at $z = 0.88$ km are broken. This means that some updrafts are deep while the other
updrafts are shallow, similar to what we have seen at $t = 8$ h. However, the mechanism that produces deep updrafts at $t = 8$ h
17 min is different from that at $t = 8$ h, as presented below. Furthermore, Fig. 14d reveals that the updrafts at the intersection
points are wider in the $x$-direction than the updrafts at other regions. This phenomenon is also seen in previous studies (e.g.,
Bai et al. 2019).

The parcels that are used to investigate the development of the third generation of convection are released at $t = 8$ h 5 min
and tracked forward for 40 min. Figure 14e shows that parcels are lifted from the regions with vertical velocity greater than 2
m s$^{-1}$. For the third generation of convection, we use a critical height of 8 km to distinguish the parcels that rise to high levels





from the parcels that do not rise to high levels. Figure 14f indicates that most of the parcels that rise above 8 km are lifted
from the two intersection points; only a small number of parcels that rise above 8 km are lifted from regions other than the
two intersection points.

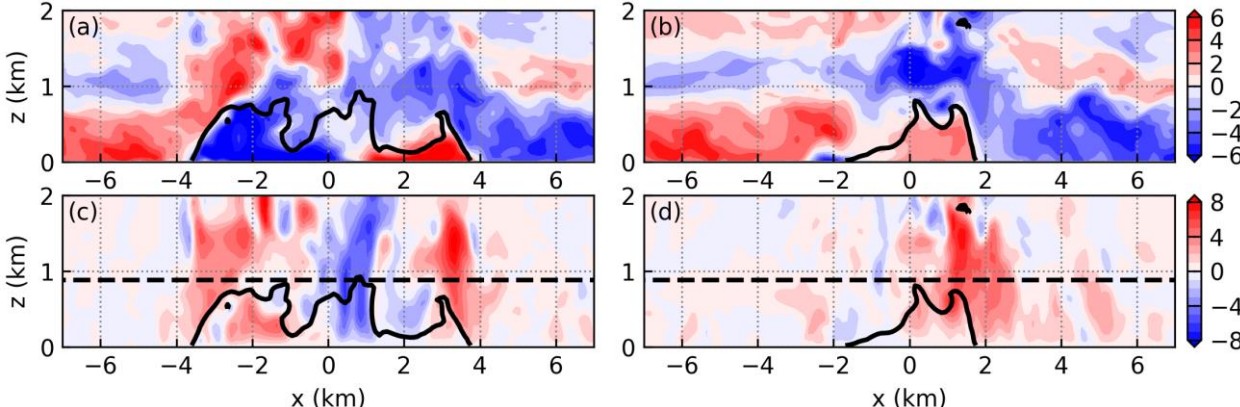

**Figure 15: Vertical cross section of (a) cross-coast wind and (c) vertical velocity at $y = 6.55$ km. (b) and (d) are the same as (a) and (c) except at $y = 0.75$ km. The black contour indicates the edge of the cold pool. The dashed line indicates the LFC. The results are**
**at $t = 8$ h 17 min of simulation THF700.**

Figures 15a and 15c respectively show the vertical cross section of cross-coast wind and vertical velocity at $y = 6.55$ km at $t$
$= 8$ h 17 min. Near their edges, the depths of the cold pool are comparable to those of the sea breezes. In this situation, it is
the collisions between the gust fronts and the sea-breeze fronts (Fig. 15a), instead of the collision of the sea-breeze fronts (cf.
12a), that produce the deep updrafts (Fig. 15c). Actually, deep updrafts are produced near almost all edges where the cold
pool is deeper than 0.4 km (cf. Figs. 14a, and 14d).

Figure 15c also shows that the left updraft is farther away from the downdraft than the right updraft. In the following 20
minutes, some of the parcels lifted by the left updraft do not encounter strong downdrafts (Fig. 14f), so the convective cells
developing from the left updraft rise above 8 km. However, the parcels that are lifted from the right updraft are quickly
entrained into the downdrafts (not shown), so the convective cells developing from the right updraft quickly dissipate.

Figures 15b and 15d show the vertical cross section at $y = 0.75$ km, which is close to the intersection point at $(x, y) = (2, 1)$
km. The cold pool depth is approximately 0.7 km near the right edge (Fig. 15b). A deep updraft is produced through the
collision of the gust front and the sea-breeze front (Fig. 15d). Figure 15d also shows that the updraft at $y = 0.75$ km is much
wider than that at $y = 6.55$ km (cf. Fig. 15c), as mentioned above. Furthermore, the parcels lifted by the deep updraft do not
encounter strong downdrafts in the following 20 minutes. Thus, the convective cells developing from the intersection points
manage to rise above 8 km.

Another factor is found to invigorate the third generation of convection. Figure 14f shows that there are multiple clusters of
parcels that rise above 8 km in the regions enclosed by the grey rectangle, indicating that multiple convective cells are
produced. Parcel trajectory analysis reveals that these convective cells merge with each other (not shown). As mentioned in





Sect. 1, the merger of convective cells reduces the detrimental effect of entrainment so the convective cells that merge rise to

higher levels than the convective cells that do not merge.

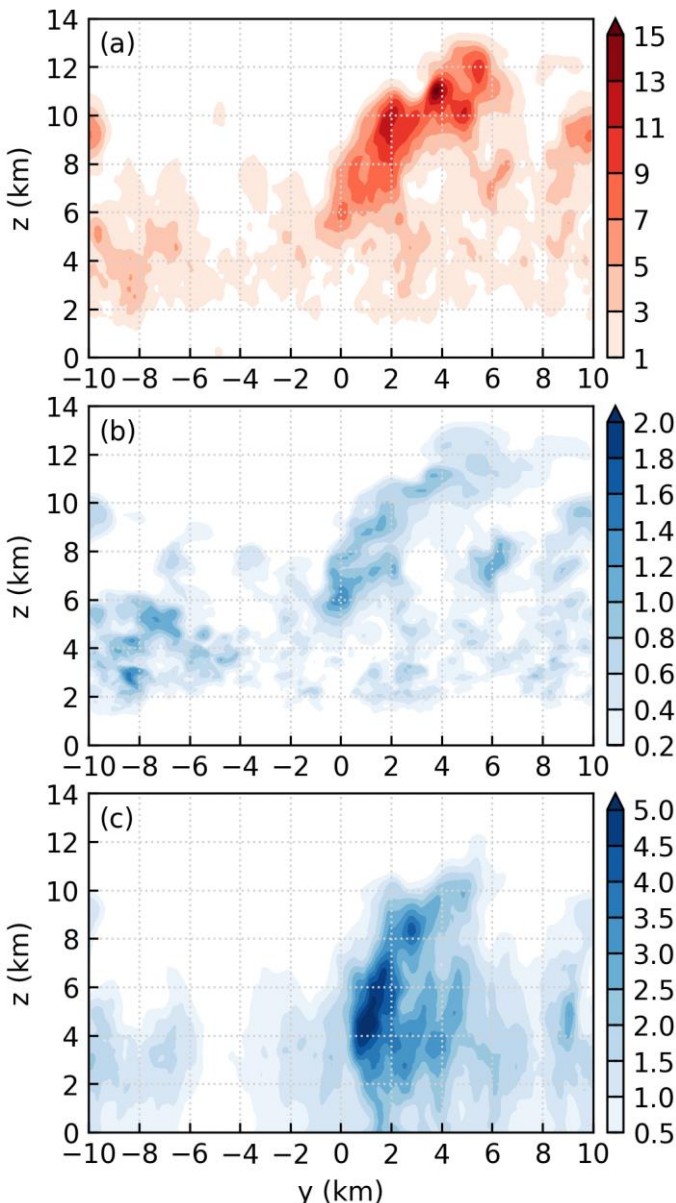

**Figure 16: The same as Fig. 10, except at $t$ = 8 h 35 min.**

We now summarize the processes through which the third generation of convection is produced. Two cold pools are produced by the second generation of convection. Near the intersection points of the cold pools, deep and wide updrafts are

produced through the collision between the gust front and the sea-breeze front. Convective cells develop from the deep and





wide updrafts near the intersection points, as well as from some other regions where the updrafts are deep and wide. Furthermore, the convective cells that are close to each other merge, consequently producing a deep convective cell. It has been shown that the updrafts producing the third generation of deep convection are larger than those producing the second generation (cf. Figs. 11 and 14). As mentioned in Section 1, clouds that are bigger at their bases rise to higher levels.

Furthermore, the merger of convective cells further invigorates the third generation of convection. As a result, the third generation of convection is stronger than the second generation.

Figures 16a-c respectively show the $y$-$z$ plots of the vertical velocity, cloud water mixing ratio, and rain water mixing ratio averaged from $x = -2$ to 2 km at $t = 8$ h 35 min. The cloud top has reached the height of 13 km. Such a strong convective cell produces even more cloud water (Fig. 16b) and rain water (Fig. 16c) than the second generation of convection.

**6 Sensitivity to total heat flux**

**Table 1: Boundary layer height ($z_i$) and the horizontal velocity scale ($U$).**

| Simulation | $z_i$ (km) | $U$ (m s$^{-1}$) |
|---|---|---|
| THF700 | 1.30 | 4.62 |
| THF500 | 1.18 | 3.89 |
| THF300 | 1.02 | 2.59 |

Before discussing the difference among the three simulations, we tabulate two important parameters in Table 1, i.e., $z_i$ and a horizontal velocity scale ($U$). The former measures the depth of the sea breeze, and the latter measures the speed of the sea breeze. Note that $z_i$ and $U$ are calculated at the last 10-min output time before the collision of sea-breeze fronts. Similar to

Antonelli and Rotunno (2007), the horizontal velocity scale is defined as $U = 0.5(U_l - U_r)$, where $U_l$ and $U_r$ are respectively the maximum speed of the $y$-averaged cross-coast wind at the left and the right coast. As expected, Table 1 indicates that the sea breeze becomes shallower and slower as total heat flux decreases.

Figure 17 shows the number, mean horizontal size, and mean temperature surplus of the convective cells in each simulation. At each height, a grid point is defined as within a convective cell if its vertical velocity is greater than a threshold of 10 m s$^{-1}$.

Tests show that using a threshold of 5 or 15 m s$^{-1}$ gives qualitatively the same results. These grid points are then four-connected to form clusters, and each cluster is defined as a convective cell. The size of the convective cell $D = \sqrt{A}$, where $A$ is the area of the convective cell. Note that the mean size of the convective cells is the size averaged over all convective cells, while the mean temperature surplus is the temperature surplus averaged over all grid points identified as within convective cells. In addition, the ellipses in Fig. 17 indicate different generations of convection.





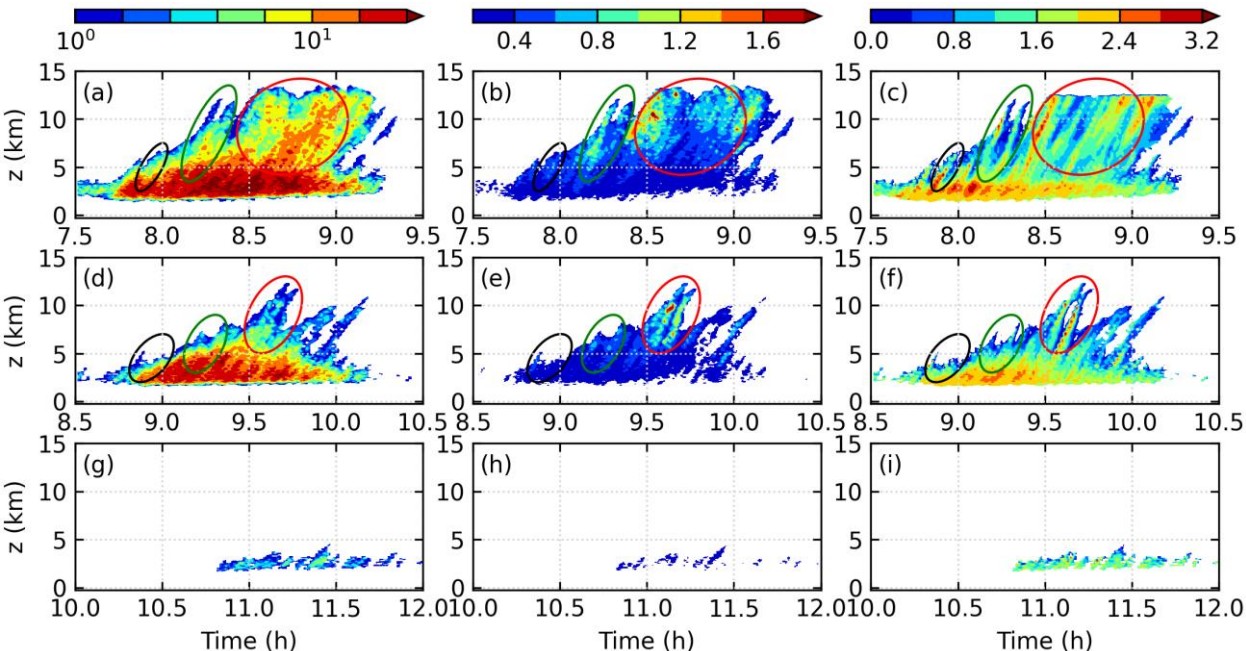

**Figure 17: (a) Number, (b) mean size (km), and (c) mean temperature surplus (K) of the convective cells in simulation THF700. Note that a logarithmic scale is used in (a). (d)-(f) and (g)-(i) are the same as (a)-(c) except in simulation THF500 and THF300. The black, green, and red ellipses indicate the first, second, and third generation of convection, respectively.**

In simulations THF700 and THF500, there are more than 10 convective cells below $z = 4$ km (Figs. 17a and 17d), and their mean size is less than 0.4 km (Figs. 17b and 17e). Above $z = 4$ km, the number of convective cells generally decreases with height (Figs. 17a and 17d), while the mean size of the convective cells is relatively constant (Figs. 17b and 17e). Two factors contribute to this phenomenon: first, the small convective cells dissipate during their ascent and only the big convective cells rise to high levels; second, mergers occur among some convective cells. In addition, the convective cells are always positively buoyant above $z = 4$ km (Figs. 17c and 17f), indicating that they are resistant to the detrimental effect of entrainment.

Figures 17a-c show that the later generation of convection is stronger than the earlier generation of convection in simulation THF700, as explained in Sect. 5. In simulation THF500, the development of the three generations of convection is generally similar to that in simulation THF700, so the later generation of convection is also stronger than the earlier generation of convection. However, because the sea breezes are shallower and slower (Table 1), the forcing of the first and the second generation of convection is weaker in simulation THF500. As a result, the first and second generation of convection in simulation THF500 are both weaker than their counterparts in simulation THF700. Furthermore, the weaker second generation of convection produces shallower cold pools. The forcing of the third generation of convection is therefore weaker. As a result, the third generation of convection in simulation THF500 is also weaker than in simulation THF700. In simulation THF300, the sea breezes are even weaker, so only a few weak convective cells are produced (Figs. 17g-i).





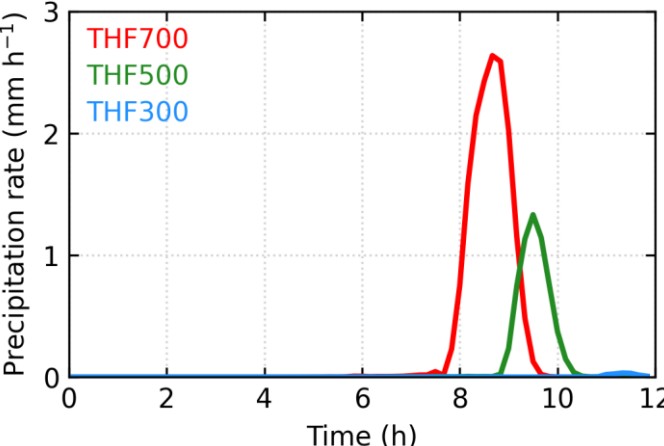


**Figure 18: Mean surface precipitation rate over the land part.**

Figure 18 shows the mean surface precipitation rates over the land part for the three simulations. In simulation THF700, the sea-breeze fronts collide at $t$ = 7 h 50 min. Before the collision, negligible surface precipitation is produced; after the collision, a large surface precipitation rate is produced as a result of the successive developments of convection. In simulation THF500, the two sea-breeze fronts collide at $t$ = 8 h 50 min, which is one hour later than that in simulation THF700. This is because decreasing sensible heat flux reduces the propagation speed of the sea-breeze fronts (Antonelli and Rotunno, 2007). Due to the weaker convection, a smaller surface precipitation rate is produced in simulation THF500 than in simulation THF700. In simulation THF300, the sensible heat flux is even smaller, so the sea-breeze fronts move even more slowly and collide at $t$ = 10 h 50 min. Negligible surface precipitation is produced in this simulation.


## 7 Summary


A series of large-eddy simulations are performed to investigate the processes involved in deep-convection initiation (DCI) over a peninsula. In each simulation, two sea-breeze circulations develop, with the two sea-breeze fronts moving inland. When the sea-breeze fronts collide near the centerline of the domain, CAPE is substantially increased due to the vapor transported by the sea breezes, and strong updrafts are produced due to the collision of sea-breeze fronts.

DCI occurs after the collision of sea-breeze fronts. In the simulations with a maximum total heat flux of 700 and 500 W m$^{-2}$, DCI occurs through the development of three generations of convection. The first generation of convection develops as a direct result of the sea-breeze-fronts collision, and the position of the first generation of convection is random. The second generation of convection is also produced mainly through the collision of sea-breeze fronts while the collision between gust fronts and sea-breeze fronts plays a minor role. The position of the second generation of convection is affected by the first generation of convection: the second generation of convection does not develop in regions where strong downdrafts are produced by the first generation of convection, but instead develops in regions where no strong downdrafts are produced.







The third generation of convection mainly develops from the intersection points of the cold pools that are produced by the second generation of convection. Parcels lifted from regions other than the intersection points also play a minor role in producing the third generation of convection. In addition, merger occurs to convective cells that are close to each other, further invigorating the third generation of convection.

As the total heat flux decreases from 700 to 500 W m$^{-2}$, both the height and speed of the sea breezes are reduced, so the forcing of the first and second generations of convection is reduced. These two generations of convection are therefore weaker. Furthermore, the weaker second generation of convection produces shallower cold pools, reducing the forcing of the third generation. Consequently, the third generation of convection is also weaker. As the total heat flux further decreases to 300 W m$^{-2}$, the sea breezes are even shallower and weaker. Only one generation of shallow convection is produced.

**Appendix A**

Before performing the compositing, we need to define the positions of sea-breeze fronts. Because the simulation setup is homogeneous in the $y$-direction, we only need to define the positions of sea-breeze fronts in the $x$-direction. First, the cross-coast wind is averaged along the $y$-direction. Then, a running average is performed to remove the effects of turbulence. By trial and error, we find that performing the running average twice to the $y$-averaged cross-coast wind gives the best result. The window for running average is 5 km. Finally, the location having the maximum horizontal convergence in the left half of the domain is defined as the position of the left sea-breeze front; the location having the maximum horizontal convergence in the right half of the domain is defined as the position of the right sea-breeze front.

We also need to define the position of each thermal. Similarly, we only define the position of a thermal in the $x$-direction. On a reference horizontal plane at $z = 0.5z_i$, any grid point with vertical velocity greater than $1.5w^*$ is defined as within a thermal. Here, the boundary layer height $z_i$ is defined as the height of the lowest grid point with $\frac{d\bar{\theta}}{dz} > 3$ K km$^{-1}$, where $\bar{\theta}$ is the potential temperature averaged from $x = -1$ to 1 km and $y = -10$ to 10 km. The convective velocity scale $w^*$ is defined as

$$w^* = \left(\frac{g}{\bar{\theta}} z_i \overline{w'\theta'}\right)^{1/3}, \text{(A1)}$$

where $g$ is the gravitational acceleration, and $\overline{w'\theta'}$ is the turbulent flux of potential temperature at the surface. All grid points that are identified as within a thermal are then four-connected in the horizontal plane to form clusters. Each cluster is defined as a thermal. The mean $x$-coordinate of all grid points within the thermal is defined as the position of the thermal. According to the distance between the thermal and the sea-breeze front, a thermal is further categorized as a left-front thermal, a right-front thermal, or an intermediate thermal, as described in the Sect. 4.1.

A procedure similar to that used by Finnigan et al. (2009) and Schmidt and Schumann (1989) is used to composite the thermals at a given output time. As an example, we detail the procedure of compositing the left-front thermals. First, for each left-front thermal, we find the grid point having the maximum vertical velocity in the plane at $z = 0.5z_i$. The position of these grid points is labeled as $(x_m, y_m)_i$, where $i$ is the index of the left-front thermals. Note that each $(x_m, y_m)_i$ indicates a

3-D thermal. Then, all the left-front thermals are shifted horizontally so that all $(x_m, y_m)_i$ coincide. A new coordinate system is therefore established where the coincident $(x_m, y_m)$ is at $(0, 0)$. Finally, ensemble averaging is performed over all the left-front thermals, giving rise to the composite left-front thermal at the output time. By repeating this procedure, we can obtain the composite left-front thermals at all output times.

We also need to average the composite left-front thermals over a certain period of time. Because the boundary layer is deepening, the thermal size tends to increase with time. In addition to the boundary layer height, the sensible heat flux is also evolving, so the updraft strength also varies with time. In this situation, the size and vertical velocity of the composite thermal are respectively nondimensionalized by $z_i$ and $w^*$ before they are averaged over time. The same procedure is used to calculate the mean composite right-front thermal and mean composite intermediate thermal.

**Code and data availability**

The CM1 model is publicly available at https://www2.mmm.ucar.edu/people/bryan/cm1/getcode.html. Please contact S. Fu for the model output data.

**Author contribution**

SF and RR designed the study. SF performed the simulations and analyzed the results. All authors commented on the results and co-wrote the paper.

**Competing interests**

The authors declare that they have no conflict of interest.

**Acknowledgements**

The research product of L1 gridded data (produced from Himawari-8) that was used in this paper was supplied by the P-Tree System, Japan Aerospace Exploration Agency (JAXA). The sounding data was downloaded from http://weather.uwyo.edu/upperair/sounding.html. ERA5 data was generated using Copernicus Climate Change Service Information [2019]. We thank George Bryan for providing and helping us with the CM1 model. This project is supported by National Science Foundation of China (41930968, 42075067, and 42005059). The visit of S. Fu to the National Center for Atmospheric Research is supported by China Scholarship Council. The National Center for Atmospheric Research is sponsored by the National Science Foundation.



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
