# Peer review of "A large-eddy simulation study of deep-convection initiation through the collision of two sea-breeze fronts"

_Atmospheric Chemistry and Physics, 2021_

## Author Response (AR1)

We thank the editor and referees for the careful reading and constructive suggestions.

Below, the referees' comments are in Black, our responses are in Blue, and the changes in the manuscript are in Red.

**Response to Daniel Kirshbaum's comments:**

Review of "A large-eddy simulation study of deep-convection initiation through the collision of two sea-breeze fronts" by Fu et al.

In this manuscript, large-eddy simulations of conditionally unstable flow over a peninsula are conducted to examine the processes leading to the initiation of deep convection. The simulations are configured to roughly represent Leizhou Peninsula of southern China, but idealized to facilitate physical interpretation. The authors conclude that the convection develops in three "generations", each deeper and more intense than the previous one. They explain this evolution through analysis of boundary-layer and cloud-layer thermals, which appear to widen and deepen over the course of the simulation. Experiments evaluating the sensitivity of the solution to the surface total heat flux (with a fixed Bowen ratio of 0.2) are also conducted.

I found the manuscript to be well written, the figures easy to understand, and the arguments clearly formulated. The simulations results are clearly presented, authors conduct some interesting and novel analyses to gain insight, and the conclusions are sensible. However, I think that some of the conclusions are weaker than others, due to questions on the underlying logic and the extent to which the evidence presented supports them. Therefore, in my major comments below, I recommend the authors do some modest additional analyses to strengthen what appear to be the weaker conclusions.

We thank the referee. Please see the point-to-point response below.

Major comments:

The manuscript explains the three generations of convection based primarily based on the widening of boundary-layer thermals over the course of the event. I have two related concerns. First, it is possible that processes in the cloud layer also play a role in the deepening of convection over the day. Does the midtropospheric humidity meaningfully change over the island (specifically over the sea-breeze convergence line) during the day? How might that affect the cloud development? Also, (ii) does the entrainment and cloud buoyancy substantially change over the three generations? Past studies have established a link between boundary-layer thermal size and cloud properties, but every problem is different and it would be useful to examine whether those same hypotheses apply here.

For the first point, we agree that the processes in the cloud layer affect the deepening of the convection. We calculated the profile of vapor mixing ratio averaged from $x = -2$ to 2 km and from $y = -10$ to 10 km for simulation THF700, as shown in Fig. 1. It is seen that the vapor mixing ratio above $z = 2$ km starts to increase after $t = 7$ h 30 min, because of the upward transport of vapor by the convective cells.

[Figure]

Figure 1: Vapor mixing ratio (g kg$^{-1}$) averaged from $x = -2$ to 2 km and from $y = -10$ to 10 km for simulation THF700.

We performed a sensitivity test to show that the increase of vapor above $z = 2$ km does promote the development of the later convective cells in simulation THF700. The

convection starts to develop after $t$ = 7 h 30 min. Therefore, we nudge the vapor mixing ratio in non-cloudy grid points after $t$ = 7 h 30 min to the mean vapor mixing ratio averaged from $x$ = -2 to 2 km and from $y$ = -10 to 10 km at $t$ = 7 h 30 min. A grid point is defined as non-cloudy if it satisfies $(q_c + q_r)$ < 0.01 g kg$^{-1}$, where $q_c$ and $q_r$ are respectively the cloud and rain water mixing ratio. The nudging is applied to non-cloudy grid points only, so that the vapor mixing ratio in cloudy grid points is affected only indirectly by entraining the drier environmental air. The nudging is applied as (Schlemmer et al., 2011)

$$\frac{\partial q_v}{\partial t} = \left(\frac{\partial q_v}{\partial t}\right)_{other\ terms} - \frac{q_v - q_{v,ref}}{\tau},$$

where $q_v$ is vapor mixing ratio, $\left(\frac{\partial q_v}{\partial t}\right)_{other\ terms}$ the tendency due to the other terms, $q_{v,ref}$ the prescribed value, and $\tau$ the time scale of nudging. We set $\tau$ to be 60 s, so that the profile remains very close to the prescribed profile. Figure 2 shows the number, size, and buoyancy for simulation THF700 and the simulation with nudging.

[Figure]

Figure 2: (a) Number, (b) mean size (km), and (c) mean temperature surplus (K; the difference between the density potential temperature of a grid point to the reference density potential temperature) of the convective cells in simulation THF700. Note that a logarithmic scale is used in (a). Panels (d)-(f) are the same as (a)-(c) but for the simulation with nudging. The black, green, and red ellipses indicate the first, second, and third generation of convection, respectively.

Examination of the 1-min results shows that there are also three generations of convection developing in the simulation with nudging. In addition, Fig. 2 shows that the

first generation of convection is of similar strength between the simulations with and without nudging; while the second and third generations in the simulation with nudging are weaker than their counterparts in the simulation without nudging. This sensitivity test suggests that the upward vapor transport by the earlier convective cells does promote the development of the later convective cells.

In the simulation with nudging, we can also see that the later generation of convection is slightly stronger than the earlier generation, in terms of the size (Fig. 2e) and the temperature surplus (Fig. 2f) of the convective cells.

In the present study, we mainly focus on the processes within the boundary layer. Thus, we think it is sufficient to briefly mention the results of the sensitivity test in the revised manuscript.

The related change in the manuscript is at line 401.

Third, the first generation of convection moistens the troposphere below $z = 6$ km. This reduces the dilution of the subsequent convective cells, and hence tends to invigorate the second generation of convection. This point is supported by a sensitivity test, where the vapor mixing ratio of the non-cloudy grid points is strongly nudged to the mean vapor mixing ratio averaged from $x = -2$ to 2 km and from $y = -10$ to 10 km at 7 h 30 min. A grid point is defined as non-cloudy if its sum of cloud water and rain water mixing ratio is less than 0.01 g kg-1. The nudging is applied only for $z > 2$ km. The sensitivity test (not shown) shows that the second generation of convection is only slightly stronger than the first generation of convection.

At line 470.

Third, the moistening effect by the first two generations of convection further promotes the development of the third generation of convection. In the sensitivity test where the vapor mixing ratio of the non-cloudy grid points is strongly nudged, the third generation of convection is only slightly stronger than the second generation of convection, which is only slightly stronger than the first generation of convection.

For the second point, we did analyze the buoyancy, as shown in Fig. 17 of the original manuscript. It is seen that the buoyancy does increase over the three generations of convection.

Traditionally, entrainment rate is calculated for a cloud ensemble. In the present study, the number of convective cells is small, so we cannot use the traditional method to calculate the entrainment rate. The recently developed method, such as that described in Romps (2010), can be used to calculate the entrainment rate of individual cloud, but is very complicated. Using such a method might be too much for the present paper. In addition, Fig. 17 of the original manuscript shows that buoyancy increases over the three generations. This shows indirectly that entrainment rate decreases over the three generations.

Reference:

Romps, D. M.: A direct measure of entrainment, J. Atmos. Sci., 67, 1908-1927, doi: 10.1175/2010JAS3371.1.

My second concern is a common one---the cause-and-effect relation between boundary-layer forcing and moist convection. The authors argue that the convection deepens due to a widening of boundary-layer thermals, but they haven't ruled out the possibility that the boundary-layer thermals grow in response to deeper convection. It's possibly a combination of both, and the contributions of each may not be easy to isolate. Boundary-layer thermals should increase in size over the course of the day even in the absence of moist processes, but the very large thermals highlighted in Figs. 12 and 15 may have widened after they initiated convection. As a cumulus deepens, its cloud-scale circulation expands, which can lead to a widening of the boundary-layer thermal(s) supporting the cloud. One way to address this question would be to track a few of the wider boundary-layer thermals from their origin to the point where their associated clouds reach the midtroposphere (say, 4 km). If the thermal is already large before the convection becomes deep, it would strengthen the authors' causal argument.

We agree that the thermals can grow in response to the convective cells, and our arguments need to be further strengthened. Since the number of thermals is small in our simulations, we decide to track the thermals manually using the 1-min output at multiple levels, instead of using a more sophisticated tracking algorithm. Below, we will examine each generation of convection in simulation THF700.

The every-1-min time-lapse figures that are used for this analysis can be downloaded from https://doi.org/10.17605/OSF.IO/JZETH

For the referee's convenience, the most-important figures are also shown below.

To begin, let's focus on the first generation of convection. Figure 3 shows the vertical velocity at 10 levels, and Fig. 4 shows the vertical velocity, cloud water mixing ratio, and rain water mixing ratio averaged from $x$ = -2 to 2 km at $t$ = 7 h 47 min in simulation THF700. At this time, we see from Fig. 4 that the first generation of convection is at its nascent stage, and is mostly below $z$ = 4 km. At $z$ = 1.2 km of Fig. 3, it is seen that the updraft between $y$ = -8 to -2 km is wider (in the $x$-direction) than the updrafts at other places.

Figures 5 and 6 are respectively the same as Figs. 3 and 4 but at $t$ = 7 h 56 min. Comparing Figs. 3 and 5, we see that the wide updraft between $y$ = -8 to -2 km has ascended from $z$ = 1.2 km in Fig. 3 to $z$ = 4.0 km in Fig. 5. The wide updraft between $y$ = -8 to -2 km at $z$ = 4.0 km in Fig. 5 corresponds to the strongest convective cell shown in Fig. 6.

Based on the above analysis, it is seen that the updraft developing into the strongest convective cell is already wider than the other updrafts before the convective cell becomes deep.

[Figure]

Figure 3: Horizontal cross section of vertical velocity at 10 levels at $t$ = 7 h 47 min in simulation THF700.

[Figure]

Figure 4: (a) Vertical velocity (m s$^{-1}$), (b) cloud water mixing ratio (g kg$^{-1}$), and (c) rain water mixing ratio (g kg$^{-1}$) averaged from $x$ = -2 to 2 km at $t$ = 7 h 47 min in simulation THF700.

[Figure]

Figure 5: The same as Fig. 3 but at $t$ = 7 h 56 min.

[Figure]

Figure 6: The same as Fig. 4 but at $t$ = 7 h 56 min.

Now, let's focus on the second generation of convection. Figures 7 and 8 show the results at $t$ = 8 h 0 min. At this time, the convective cells of the second generation are mostly below $z$ = 4.0 km (Fig. 8). The cells above $z$ = 4.0 km are the remnants of the first generation of convection. From Fig. 7, we clearly see four thermals at $z$ = 0.8 and 1.2 km. These four thermals are of similar width and are produced through the collision of the two sea-breeze fronts.

Figures 9 and 10 show the results at $t$ = 8 h 9 min. The four thermals have ascended to higher levels and can be seen at $z$ = 2.4 to 4.0 km in Fig. 9. These thermals correspond to the four convective cells shown in Fig. 10.

From the above analysis, we see that the thermals developing into the strong convective cells are already wide before the convective cell becomes deep. In addition, since the four thermals are of similar width, we do not see that "convective cells developing from **wider** updrafts rise to **higher** levels", we only see that "**deep** convective cells develop from **wide** updrafts".

[Figure]

Figure 7: The same as Fig. 3 but at $t$ = 8 h 0 min.

[Figure]

Figure 8: The same as Fig. 4 but at $t$ = 8 h 0 min.

[Figure]

Figure 9: The same as Fig. 3 but at $t = 8$ h 9 min.

[Figure]

Figure 10: The same as Fig. 4 but at $t$ = 8 h 9 min.

Let's then focus on the third generation of convection. Figures 11 and 12 show the results at $t$ = 8 h 13 min. Figure 12 shows that the third generation of convection, which is between $x$ = 0 to 3 km, is generally below $z$ = 4.0 km; the cells above $z$ = 4.0 km belong to the second generation of convection. From Fig. 11, we see wide thermals around (x, y) = (1, 1) km at $z$ = 0.8 and 1.2 km. Actually, we can also see wide thermals at other places.

At $t$ = 8 h 22 min, the thermals around (x, y) = (1, 1) km at $t$ = 8 h 13 min has ascended to higher levels, and are between $y$ = -1 to 5 km at $z$ = 4.0 km in Fig. 13. These thermals produce the convective cell between $y$ = -1 to 5 km in Fig. 14.

Again, we see that the thermals developing into the strong convective cells are already wide before they become deep. Note that some wide thermals at $t$ = 8 h 13 min fail to develop into deep convection because they are suppressed by the downdrafts produced by earlier convective cells, as discussed in the original manuscript. Similar to the second generation of convection, we only see that "**deep** convective cells develop from **wide** updrafts".

[Figure]

Figure 11: The same as Fig. 3 but at $t$ = 8 h 13 min.

[Figure]

Figure 12: The same as Fig. 4 but at $t$ = 8 h 13 min.

[Figure]

Figure 13: The same as Fig. 3 but at $t$ = 8 h 22 min.

[Figure]

Figure 14: The same as Fig. 4 but at $t = 8$ h 22 min.

Finally, it is worth mentioning that the above conclusion can actually be supported by Figs. 9, 11, and 14 in the original manuscript. In those figures, the positions where the parcels are lifted are shown at times before the convective cells become deep, and we can see that the thermals are wide.

The related change in the manuscript is at line 323.

It is worth mentioning that at the time of Fig. 9, the convective cells are still below z = 4 km (see supplement), so the wide updraft is mostly produced by the boundary-layer forcing, and has not been substantially affected by the convective cell.

At line 352.

Note that at the time for Fig. 11, the convective cells of the second generation are mostly below z = 4 km (see supplement), so the wide updrafts shown in Fig. 11d are also mostly produced by the boundary-layer forcing, and are not substantially affected by the convective cells, similar to the first generation of convection.

At line 414.

Figures 14a and 14b respectively show the depth and the temperature surplus of the cold pool at t = 8 h 17 min, when the cloud top of the third generation is near z = 4 km (see supplement).

Minor comments

P. 5, L.~120: Why use the Kessler scheme for a simulation of deep convection? Could this be one of the reasons why deep convection fails to initiate when you use realistic total heat fluxes? I think it would be worthwhile to perform another simulation in the intermediate-heat-flux case with an ice scheme, to determine (i) if the same ideas carry over to a more realistic cloud representation and (ii) whether the lack of ice phase is the reason convection struggles to deepen in the 500 W/m^2 case.

We should clarify that deep convection is produced in both simulations THF700 and THF500 (see Fig. 17 of the original manuscript). The only observation that shows the strength of the convection is the cloud top height retrieved from the satellite data, which is about 12 km (Fig. 1e of the original manuscript). Simulations THF700 and THF500 both produce convective cells that are deeper than 12 km and are therefore consistent with the satellite observation. In this regard, deep convection does not fail to initiate in simulation THF500.

By stating "a maximum total heat flux of 700 W m$^{-2}$ is required for the model to reproduce the observations" in the original text, we mean that a maximum total heat flux of 700 W m$^{-2}$ is required to reproduce the observed **time of DCI**, instead of the strength of the deep convection.

In our configuration, the time of DCI is determined by when the two sea-breeze fronts collide. Since the movement of the sea-breeze front is not related to the microphysics, we believe that using a more realistic microphysics scheme will not change the time of DCI.

The related change in the manuscript is at line 175.

However, a maximum total heat flux of 700 W m$^{-2}$ is required for the model to reproduce the observed time of DCI.

We do believe that using a more realistic microphysics scheme changes the evolution of the convective system, so we used the Morrison two-moment microphysics scheme to rerun simulation THF500. The results are shown in Fig. 15. A comparison of Figs. 15c and 15f indicates that the peak strength in the simulation with the Morrison scheme is stronger than that in the simulation with the Kessler scheme. This is due to the slower development of precipitation, which postpones the development of downdrafts; and the extra latent heating from freezing processes, which increases the buoyancy of the clouds. Another difference is that there are three generations of convection in the simulation with the Kessler scheme, while only two generations of convection in the simulation with the Morrison scheme. This is probably because CAPE has been consumed by the end of the second generation.

[Figure]

Figure 15: (a) Number, (b) mean size (km), and (c) mean temperature surplus (K) of the convective cells in the simulation THF500 run with the Kessler scheme. Note that a logarithmic scale is used in (a). (d)-(f) are the same as (a)-(c) but for the simulation with the Morrison microphysics. The black, green, and red ellipses indicate the first, second, and third generation of convection, respectively.

[Figure]

Figure 16: (a) Vertical velocity (m s⁻¹), (b) liquid water (cloud and rain) mixing ratio (g kg⁻¹), and (c) solid water (ice, snow, and graupel) mixing ratio (g kg⁻¹) averaged from $x = -2$ to 2 km at $t = 9$ h 12 min in the simulation with a maximum total heat flux of 500 W m⁻² and the Morrison scheme.

Figure 16 shows the results of the simulation with the Morrison scheme at $t = 9$ h 12 min. Note that no solid water has been produced at this time, but will be produced in the later times (Fig. 16c). Rain shafts are produced from $x = -8$ to -6 km and from $x = 2$ to 4 km (Fig. 16b). They are produced by the first generation of convection. The convective cells shown in Fig. 16a belong to the second generation of convection. Obviously, they only develop in the regions where no downdrafts (associated with the two rain shafts)

are produced by the first generation of convection. In addition, the second generation of convection is also forced mainly by the collision of the two sea-breeze fronts (not shown). Thus, the development of the first two generations of convection in the simulation with the Morrison scheme is similar to that in the simulation with the Kessler scheme.

It is well known that the microphysics scheme is associated with substantial uncertainty. In addition, we want to mention that the droplet concentration used in the Morrison scheme is 100 $cm^{-3}$, which is an intermediate value between the polluted situation and the clean situation. Note that we do not have any information on the droplet concentration of the selected case. If the air mass affecting Leizhou peninsula is mainly of marine origin, then the droplet concentration can be substantially lower than 100 $cm^{-3}$. In this situation, the precipitation might develop fast enough, and the behavior of the Morrison scheme might be similar to the Kessler scheme.

Using the more complicated Morrison scheme substantially increases the computational burden and the disk space used to store data. In addition, the effect of microphysics on the convection is very complicated and not the focus of this study. Thus, we only briefly mention the results of the sensitivity test.

The sensitivity test using the Morrison microphysics scheme is briefly discussed in an Appendix.

Appendix A

Simulation THF500 is rerun using the Morrison two-moment microphysics scheme. We perform a sensitivity test on simulation THF500 instead of THF700, because the maximum total heat flux over the peninsula is approximately 500 W $m^{-2}$ on the chosen day. There are two major differences between the simulation using the Morrison scheme and that using the Kessler scheme. The first difference is that the peak strength of convection in the simulation using the Morrison scheme is stronger than that in the simulation using the Kessler scheme. This is due to the slower development of precipitation, which postpones the development of downdrafts; and the extra heating

from the freezing processes, which increases the buoyancy of the clouds. The second difference is that there are only two generations of convection in the simulation using the Morrison scheme, while there are three generations of convection in the simulation using the Kessler scheme. This is probably because CAPE has been consumed by the end of the stronger second generation in the simulation using the Morrison scheme. Nevertheless, detailed analysis shows that the development of the two generations of convection in the simulation using the Morrison scheme is similar to the first two generations of convection in the simulation using the Kessler scheme.

Another reason your experiment might fail to produce sufficiently deep convection with a more realistic heat flux is that you are neglecting large-scale forcing for ascent, which is often significant in real-world deep convection events. With that said, I'm not sure if the case of interest exhibited any large-scale ascent.

As mentioned above, deep convection does develop in simulation THF500, and the main point here is the time of DCI instead of the strength of the deep convection. Since the movement of the sea-breeze front is mostly not related to the large-scale ascent, the time of DCI is mostly not related to the large-scale ascent, either.

[Figure]

Figure 17: Vertical velocity (Pa s⁻¹; positive value indicates descent and negative value indicates ascent) averaged over an area of 220 km × 220 km around the Leizhou Peninsula on the selected day. The data is from ERA5 reanalysis.

We calculated the large-scale vertical velocity (in pressure coordinate) from the ERA5 reanalysis data, as shown in Fig. 17. Large-scale ascent exists only below $z$ = 2 km; while large-scale descent exists from $z$ = 2 to 7 km. Below $z$ = 2 km, the effect of large-scale ascent is probably masked by the effect of sea breezes. Above $z$ = 2 km, the effect of large-scale descent is to suppress the development of convection.

Nevertheless, as an idealized study, we think it is acceptable to neglect the large-scale vertical velocity.

P. 13, L.~285: The word "grid" instead of "grid point" is used. This error reappears in numerous places, so please fix throughout.

We have fixed this error throughout the manuscript.

P. 16, L.~325: I'm confused by "N is the number of grids averaged". Beyond the repetition of the error noted above, it is unclear whether N includes the points where w <= 0. If not, it is a conditional average.

The related change in the manuscript is at line 328.

$N$ is the number of grid points between $x$ = -2 to 2 km, either with $w$ > 0 or ≤ 0, so the number of grid points going into the averaging procedure is fixed.

P. 16, L.~330: Calling the strongest cell in the first generation to be "randomly" produced is a bit misleading. In a global sense, it is not random because it forms along a highly localized mesoscale feature (a sea-breeze convergence line). But its location *along that line* is largely random.

The related change in the manuscript is at line 337.

and its position along the colliding sea-breeze fronts seems to be random.

The explanation of the third generation of convection does not make complete sense to me. There is a discussion of how the collision of cold pools gives rise to wider boundary-layer thermals, but the authors also state on P. 21 that it is the collision between gust fronts and the sea-breeze fronts that invigorates the convection. These seem like two different arguments. Is it fair to say that the local juxtaposition of sea-breeze convergence *and* cold-pool convergence leads to the widest updrafts? That would seem to encompass both arguments.

Although we cannot exclude the possibility that the collision of cold pools also plays a role in the formation of the third generation of convection, we are sure that it is not important. The air in the cold pool is denser than the air outside the cold pool, so the collision of gust fronts cannot lift the air in the cold pool to high levels. The main role of the collision of gust fronts is to form the intersection point, where a wide updraft is formed.

For the third generation of convection, the updrafts are produced through the collision between sea-breeze fronts and gust fronts, as explained in the original manuscript. This is because the air in the sea breezes is lighter and is easily lifted.

In the third generation, the authors point to a merging of moist updrafts to generate larger clouds. While the process sounds plausible, I wonder why this only occurs in the 3rd generation and not the other two? If the merging only occurs in the third generation and not the prior two generations, the authors should attempt to explain why.

For the first generation of convection, merger does not occur because the convective cells are sufficiently separated.

For the second generation of convection, detailed analysis indicates that the two convective cells between $x$ = 2 to 10 km might have mutual influence, such as moistening the environment for each other; but they do not merge completely.

The related change in the manuscript is at line 399.

Second, the two convective cells between x = 2 and 10 km are very close to each other (Fig. 13a). They might invigorate each other by moistening the environment, as suggested by Böing et al. (2012).

P. 23, L.~460: The authors claim that, "The former [z_i] measures the depth of the sea breeze". My understanding is that z_i is the boundary-layer depth, not the sea-breeze depth. I realize that the two tend to scale together, but without that background understanding, the text seems misleading.

The related change in the manuscript is at line 477.

The first is $z_i$, which is known to scale with the depth of the sea breeze (Antonelli and Rotunno, 2007).

I don't see a justification for the statement on P. 24 L.~475 that, "Above (ð• ' §) = 4 km, the number of convective cells generally decreases with height (Figs. 17a and 17d), while the mean size of the convective cells is relatively constant (Figs. 17b and 17e)." Looking at the middle column of Fig. 17, it appears that the cloud size does increase with height.

The related change in the manuscript is at line 492.

Above $z$ = 4 km, the number of convective cells generally decreases with height (Figs. 17a and 17d), while the mean size of the convective cells increases with height (Figs. 17b and 17e).

I also question the statement on P. 25, L.~505 that, "When the sea-breeze fronts collide near the centerline of the domain, CAPE is substantially increased due to the vapor transported by the sea breezes, and strong updrafts are produced due to the collision of sea-breeze fronts.". I agree that the vapor transport is important, but what about the impact of the updrafts on the CAPE itself? They are likely to increase it by moisture convergence as well as generating some adiabatic cooling above the boundary-layer top.

CAPE can be calculated as $\int_{LFC}^{EL} g \frac{\Theta_\rho - \overline{\Theta_\rho}}{\overline{\Theta_\rho}} dz$, where $\Theta_\rho$ is the density potential temperature of the parcel and $\overline{\Theta_\rho}$ is the density potential temperature of the environment. Since the impact of updrafts mainly occurs above the boundary-layer top, it will not affect $\Theta_\rho$ but only affect $\overline{\Theta_\rho}$. In order to show how $\overline{\Theta_\rho}$ is changed, we calculated the potential temperature and vapor mixing ratio averaged from $x = $ -1 to 1 km and from $y = $ -10 to 10 km, as shown in Fig. 18.

From $t = 0$ h to 7 h 50 min, when the two sea-breeze fronts collide, we see that the impact of updrafts slightly decreases the potential temperature above the boundary-layer top (Fig. 18a), and substantially increases the vapor mixing ratio (Fig. 18b). The changes in potential temperature and vapor mixing ratio translate to changes in $\overline{\Theta_\rho}$, which are shown in Figs. 18c and 18d. It is seen that the decrease of potential temperature partly cancels the effect of the increase of vapor mixing ratio, so the net impact of updrafts on $\overline{\Theta_\rho}$ is very small. In other words, the impact of updrafts has a negligible effect on CAPE.

The main effect of updrafts on the later convection should be the moistening of the environment, as pointed out by the referee and discussed above. However, this effect cannot be seen in the calculation of CAPE.

[Figure]

Figure 18: (a) and (b) are respectively the potential temperature and vapor mixing ratio averaged from $x$ = -1 to 1 km and the whole $y$ dimension. (c) and (d) are the change of density potential temperature due to change in potential temperature and vapor mixing ratio, respectively.

We did find a point that was not mentioned in the original manuscript. From Fig. 18a, we see that the temperature is substantially increased below $z$ = 1 km, which also increases CAPE. In the original manuscript, we only emphasize that the increase of vapor mixing ratio is important.

The related change in the manuscript is at line 524.

CAPE is substantially increased due to the vapor transported by the sea breezes and the surface heating.

Appendix A, P. 26, L.~525: Here the authors mention the "maximum horizontal convergence", but don't specify the level at which this convergence is calculated.

The related change in the manuscript is at line 559.

First, the cross-coast wind at the lowest level ($z$ = 0.02 km) is averaged along the y-direction.

**Response to anonymous referee 2:**

GENERAL COMMENTS

In this paper a series of large-eddy simulations are performed to investigate the processes involved with the initiation of deep-convection regime. LES describing the collision of the two sea-breeze system in a peninsula is not a original argument and are not mentioned. Anyway this study presents a novel configuration for the numerical models that considers a cloud model joined with an offline lagrangian model.

We thank the referee.

We should point out that we used an **online** Lagrangian model, instead of an offline Lagrangian model. Please see line 149 of the original manuscript.

We thank the referee for pointing out that there are studies focusing on the LES of the collision of sea-breeze fronts. We found one such study, which is by Rizza et al. (2015). In addition, we think it appropriate to add some references to LES studies of the sea-breeze circulation in the Introduction.

The related change in the manuscript is at line 76.

Other studies have shown that large-eddy simulations (LESs) could realistically simulate the small-scale processes occurring in either a single sea-breeze circulation (Antonelli and Rotunno, 2007; Crosman and Horel, 2012) or two colliding sea-breeze circulations (Rizza et al., 2015). However, those studies did not investigate the processes of DCI.

The principal conclusion of this investigation is that DCI occurs after the collision of sea-breeze fronts through the development of three generations of convection. In this context authors gave full explanation in the manuscripts and detailed the processes involved in the three stages of convection development inside the peninsula.

From the technical point of view the analysis of data is coherent with the objective, so I recommend publication with minor revision.

We thank the referee for the positive comments.

SPECIFIC COMMENTS

In my opinion the abstract should be more general and, in some way, reformulated. It actually contains the main hypothesis (lines 15-17): "The two sea-breeze fronts move inland and collide, producing strong instability and strong updrafts near the centerline of the domain, and consequently leading to DCI". But starting from line 17 it contains the results of simulations, that is the discussion about the three stages and the intensity of surface heat flux necessary to generate a DCI. I would expect more details in the physical process being investigated and the analysis of LES runs for the conclusion final paragraph.

We thank the reviewer for the suggestion. We have rewritten the abstract.

Deep convection plays important roles in producing severe weather and regulating the large-scale circulation. However, deep-convection initiation (DCI), which determines when and where deep convection develops, has not yet been fully understood. Here, large-eddy simulations are performed to investigate the detailed processes of DCI, which occurs through the collision of two sea-breeze fronts developing over a peninsula. In the simulation with a maximum total heat flux over land of 700 or 500 W m-2, DCI is accomplished through the development of three generations of convection. The first generation of convection is randomly produced along the colliding sea-breeze fronts. The second generation of convection only develops in regions where no strong downdrafts are produced by the first generation of convection and is also mainly produced through the collision of the sea-breeze fronts. The third generation of convection mainly develops from the intersection points of the cold pools produced by the second generation of convection and is produced through the collision between the gust fronts and the sea-breeze fronts. Decreasing the maximum total heat flux from 700

to 500 W m-2 weakens each generation of convection. Further decreasing the maximum total heat flux to 300 W m-2 leads to only one generation of shallow convection.

Another point not sufficiently discussed concern the role of "environmental wind", perhaps a further run with this wind > 0 would be necessary. It is only said that "the environment wind is set to zero to simplify the analysis." But this is not sufficient in my opinion.

We completely agree that the environmental wind plays an important role in determining the DCI. However, previous coarse-resolution studies indicate that the impact of environmental wind on DCI is quite complicated. It might be too much to include the effect of environmental wind in the present study. Thus, we focus on the situation without environmental wind in the current study and leave the impact of environmental wind to a future study.

Line 135: Please provide references for ERA5 reanalysis

We have added the reference.

1 : This is not clear, above is said that "The prescription of the surface fluxes is guided by the ERA5 reanalysis data." then total heat flux is prescribed by the sinusoidal eq.1.

ERA5 reanalysis data was used as a guidance, only. The surface fluxes that are used in the model have similar variations to the ERA5 reanalysis data but are smoothed (idealized), to facilitate the interpretation of the results.

The related change in the manuscript is at line 137.

Idealized surface fluxes are prescribed based on the ERA5 reanalysis data (Hersbach et al., 2020).

At line 165 it is written that "cloud water mixing ratio greater than 0.01 g kg-1 ", but this quantity is not described in the model-setup paragraph 2.2.

The related change in the manuscript is at line 121.

Cloud microphysics is represented with the one-moment Kessler scheme, which predicts the cloud water mixing ratio and the rain water mixing ratio.

Analysis are performed each dt=10 min first then at 1 min interval, but the time step of the cloud model is not introduced.

The related change in the manuscript is at line 136.

An adaptive time step is used, and the longest time step is 2 s.

The "connection" between the cloud and lagrangian model is realized every time step ??? The description from line 149 to 155 is quite superficial, other studies doing this coupling should be mentioned.

As mentioned above, we actually used an **online** Lagrangian model, so the positions of the Lagrangian parcels are updated at every time step.

The related change in the manuscript is at line 151.

The resolved-scale velocity is linearly interpolated to the position of the parcel, and the second-order Runge-Kutta method is used to update the position of the parcel. A similar method is used by Yang et al. (2015).